

# Low temperature quantum bounds on simple models

**Silvia Pappalardi⋆ and Jorge Kurchan**

Laboratoire de Physique de l'École normale supérieure, ENS, Université PSL, CNRS,
Sorbonne Université, Université de Paris, F-75005 Paris, France

⋆ silvia.pappalardi@phys.ens.fr

## Abstract

In the past few years, there has been considerable activity around a set of quantum bounds on transport coefficients (viscosity) and chaos (Lyapunov exponent), relevant at low temperatures. The interest comes from the fact that Black-Hole models seem to saturate all of them. The goal of this work is to gain physical intuition about the quantum mechanisms that enforce these bounds on simple models. To this aim, we consider classical and quantum free dynamics on curved manifolds. These systems exhibit chaos up to the lowest temperatures and – as we discuss – they violate the bounds in the classical limit. First of all, we show that the quantum dimensionless viscosity and the Lyapunov exponent only depend on the de Broglie length and a geometric length-scale, thus establishing the scale at which quantum effects become relevant. Then, we focus on the bound on the Lyapunov exponent and identify three different ways in which quantum effects arise in practice. We illustrate our findings on a toy model given by the surface of constant negative curvature — a paradigmatic model of quantum chaos — glued to a cylinder. By exact solution and numerical investigations, we show how the chaotic behaviour is limited by the quantum effects of the curvature itself. Interestingly, we find that at the lowest energies the bound to chaos is dominated by the longest length scales, and it is therefore a collective effect.

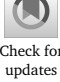

# 1  Introduction

Since the early days of quantum mechanics, physicists have tried to understand the different ways in which it affects the familiar, classical, world. In the past few years, there has been a renewed interest in the constraints posed by quantum effects on the physical properties of macroscopic systems at low temperatures. Initially, these concerned anomalous transport properties, like the *conductivity* [1], the resistivity [2] or the *viscosity* $\eta$ in the Kovtun-Son-Starinets bound [3]

$$\frac{\eta}{S} \geq \frac{\hbar}{4\pi} \, , \tag{1}$$

where $S$ is the volume density of entropy. (In our units $k_B = 1$). Even if there is no general consensus on the bounds on transport, it was argued that they are related to the emergence of

a Planckian time-scale $\tau_{\mathrm{Pl}} = \hbar/T$, depending only on the Planck constant $\hbar$ and on the absolute temperature $T$ in units of energy [4]. Recently, such Planckian time-scale has appeared in a different bound: the one on the *quantum Lyapunov exponent* $\lambda$, defined as the growth rate of some quantum out-of-time ordered correlator (OTOC) [5]. In Ref. [6], Maldacena, Shenker and Stanford proved that the rate $\lambda$ of a closely related regulated OTOC shall be bounded by[1]

$$\lambda \leq \frac{\pi T}{\hbar} \ . \tag{2}$$

This result is now known as *the bound to chaos*. Intriguingly, models of black holes, including the Sachdev-Ye-Kitaev (SYK) [7] model, saturate these bounds. Spurred by the Field-Theory/Black-Hole community, these topics have now spread over different fields, ranging from quantum Information Theory to Condensed Matter and Statistical Physics. These studies have established that quantum Lyapunov exponents are defined only for systems of $N$ elementary constituents (such as spins or fermionic sites) with all-to-all interactions in the large $N$ limit [7], or for an underlying semiclassical chaotic limit [8]. On the other hand, the bound to chaos corresponds to a genuine quantum effect.

If one considers the bounds themselves – and not the properties of systems saturating them – one may suppose that there is the quantum Uncertainty Principle at the bottom. From this down-to-earth perspective, it is a question of elementary physics to understand how quantum effects act to enforce the bounds in practice.

The goal of this work is to learn as much as possible with the simplest models we can construct: the classical and quantum free dynamics on curved manifolds. Let us now motivate this choice. To begin with, we restrict ourselves to bosonic systems in the semi-classical limit at low-temperature $T$. For the bounds to be effective one needs $T/\hbar$ to be finite, which holds in the semiclassical limit for very low temperatures corresponding to the lowest classical energies of the system for $T > 0$. Secondly, we wish to restrict to systems that have non-trivial, "interesting", dynamics down to the lowest temperatures. For instance, in an isolated minimum of the Hamiltonian, at low temperatures, the quantum system will only perform vibrations with quantum fluctuations (elementary excitations/quasi-particles) around the classical ground state. Bounds will be then satisfied, but trivially, i.e. due to the linearization of the dynamics. On the other hand, in the presence of ground-state degeneracies, or quasi-degeneracies at the lowest energies, the system may instead be non-harmonic (and chaotic) even "at the bottom of the well" [9].

With this in mind, there are two natural choices for nontrivial low-temperature dynamics: free propagation bounded by walls – *billiards* – or on *curved manifolds*. Billiards and free motion on manifolds share a scale-invariance property: the configuration-space trajectories are given, and, at different energies, the system just runs through them at different speeds. As we shall see, a billiard is nothing but a "deflated" manifold [10]. For this reason, we concentrate on the free dynamics of a particle of mass $\mu$ on a curved manifold. In particular, we focus on the example of the pseudosphere, i.e. the *surface with constant negative curvature*, a paradigmatic model of quantum chaos [11]. Far from being only of abstract theoretical interest, lattices on hyperbolic geometry have been recently experimentally realized via superconducting coplanar waveguide resonators [12]. This has paved the way for the study of exciting phenomena, that can now be experimentally probed [13].

Our ultimate motivation, however, is the possible extension of our results to *macroscopic systems*, wherein one can define a thermodynamic limit. As we will argue, explicit examples of

---

[1]The bound to chaos is usually stated as $\lambda' \leq 2\pi T/\hbar$ [6]. The missing factor "2" in Eq.(2) comes from the different definition of the exponential growth from the square commutator [cf. Eq.(26)], i.e. $\lambda' = 2\lambda$. We make this choice to simplify the comparisons with the classical Lyapunov exponent.

macroscopic billiards or macroscopic manifolds are given by hard-sphere systems [14] or by certain spin-liquids [15] respectively. For instance, the scaling of the classical $\lambda$ predicted on manifolds has been found numerically in a spin liquid in Ref. [15].

The results of this paper can be divided into two main parts. First of all, we discuss generic properties of classical and quantum free systems on Riemannian manifolds which can be derived by pure dimensional analysis. We show that these systems are automatically Planckian, i.e. characterized by a time scale $\tau_{\mathrm{Pl}}$. We illustrate how the quantities of interest for the bounds (viscosity and Lyapunov exponent) are only functions of the ratio between the smallest length-scale of the model and the thermal de Broglie wavelength. Remarkably, the classical limit of these quantities reveals clear violations of the quantum inequalities (1)-(2), as illustrated pictorially in Fig.1. This leads us to our second main scope: we focus on the bound to chaos (2) and we provide a thorough discussion of the different ways quantum mechanics intervenes to enforce it. These are discussed in full detail on a simple toy model on a two-dimensional surface that we can solve exactly and simulate numerically. One of the suggestions of our discussion is that, for a system to approach the bound for $T \to 0$, there has to be a full hierarchy of length scales, such that at each temperature some degrees of freedom are still classical, and some are on the verge of being affected by quantum effects. All these effects may be extended to high-dimensional manifolds, that are directly related to macroscopic models.

These findings open up a novel perspective on the quantum bounds, sometimes believed to be only a subject of high-energy physics [1,3,6]. Instead, they can be understood as a non-trivial effect of quantum uncertainty, even on simple models. Our framework calls for a more general study of many-body systems whose ground-state classical configurations are curved manifolds embedded in phase space.

The rest of the paper is organized as follows. Sec.2 contains a summary of the main results of the paper. Then, we first briefly comment on the relation between billiards and collapsed manifolds in Sec.3. In Sec.4, we introduce the classical and quantum dynamics on Riemannian manifolds and perform the dimensional analysis leading to the universal dependence of the viscosity and Lyapunov exponent. We also discuss how the semiclassical approximation automatically displays violations of the bounds. We then focus only on the bound to chaos in Eq.(2). In Sec.5, we explore quantum effects on simple models on two-dimensional surfaces, that we can solve exactly and simulate numerically. In Sec.6 we discuss how the bound to the Lyapunov exponent holds in the limit of zero temperature in presence of a hierarchy of length scales. We discuss generalizations to macroscopic $N$-dimensional systems in Sec.7. We conclude in Sec.8 with some closing remarks and perspectives for future work. This paper aims at being pedagogical. For this reason, all the details of the calculations are reported in the appendix.

## 2 Summary of the results

In this work, we study the bounds (1)-(2) analyzing the quantum dynamics of a free particle on a curved manifold. Besides the Planck constant $\hbar$, these systems are characterized by only few other parameters: the mass $\mu$, the kinetic energy density – that we take as thermal $\frac{T}{2}$ – and the geometry of the manifold, that we describe in terms of some characteristic length-scale $R$ and other dimensionless parameters $\vec{\alpha}$ [2]. Quantum effects become dominant depending on

---

[2]For instance all the other length-scales in units of $R$.

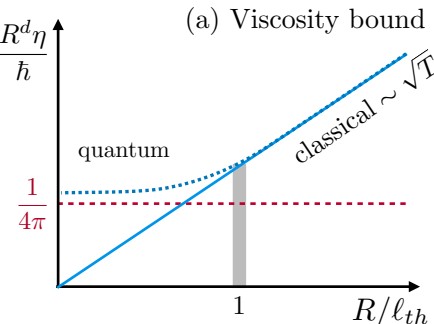
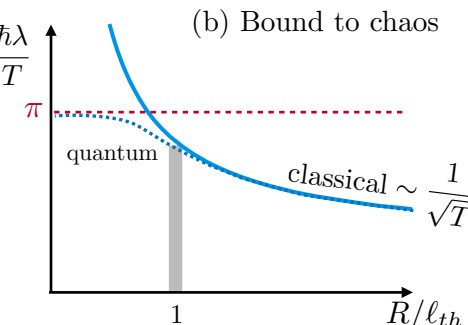

Figure 1: General picture of the quantum bounds on free manifolds. (a) The quantum dimensionless viscosity and (b) Lyapunov exponent are only a function of $R/\ell_{\mathrm{th}}$. The classical limit (blue full line) violates the bounds (1)-(2) (red dashed line) for $R < \ell_{\mathrm{th}}$. At that length-scale, quantum effects enter the game implementing the bounds.

the comparison between $R$ with the *thermal de Broglie length* [16]

$$\ell_{\mathrm{th}} = \sqrt{\frac{2\pi\hbar^2}{\mu\, T}}\,. \tag{3}$$

The content of this work can be summarized as follows:

- Quantum free systems on manifolds are naturally *Planckian*, since they are characterized only by the universal time-scale $\tau_{\mathrm{Pl}} = \hbar/T$.

- The quantities of interest for the bounds (viscosity and Lyapunov exponent) are universal functions of $R$ and the $\ell_{\mathrm{th}}$, as summarized in Fig.1. By dimensional analysis, we find

$$\frac{\eta R^d}{\hbar} = \Omega_\eta\left(\frac{R}{\ell_{\mathrm{th}}}\right)\,, \qquad \frac{\lambda\hbar}{T} = \Omega_\lambda\left(\frac{R}{\ell_{\mathrm{th}}}\right)\,. \tag{4}$$

The $\Omega_{\eta/\lambda}(x)$ are dimensionless functions that only depend on the geometry of the problem. In the classical limit $x = R/\ell_{\mathrm{th}} \gg 1$, their leading behaviour is

$$\Omega_\eta(x) \simeq x + \mathcal{O}(x^{-2})\,, \qquad \Omega_\lambda(x) \simeq \frac{1}{x} + \mathcal{O}(x^{-2})\,, \tag{5}$$

which displays clear violations of the bounds (1)-(2) when $x = R/\ell_{\mathrm{th}} \to 0$, as illustrated in Fig.1. Therefore, the quantum effects shall act when $\ell_{\mathrm{th}} \sim R$.

- We restrict to the bound to chaos in Eq.(2) and at this scale we identify three mechanisms enforcing it:

  (1) (trivial) *finite size effects*: $\ell_{\mathrm{th}}$ is of the order of the system size, and one can not perform any semi-classical description.

  (2) In the presence of different negative curvatures, the bounds are implemented by an *effective repulsive potential*, which at low energies excludes the particle from the most curved and chaotic regions. In this framework, the bounds are saturated at zero temperature in the presence of multiple (diverging) length-scales and are hence a collective effect.

(3) In the presence of only one negative curvature, there is a regime where one finds, surprisingly, a super-exponential growth of OTOCs, which makes the Lyapunov regime - thus the bound in Eq.(2) - ill defined.

These effects are studied in detail for a toy model with constant and negative curvature, that we solve and study numerically.

## 3 Billiards are "collapsed" manifolds

Before starting our discussion, we shall make a brief remark. As was pointed out by Arnold, a billiard may be seen – both classically and quantum mechanically – as a particular case of a manifold [10]. Thus, an ellipsoid in the deflated limit becomes an ellipsoidal billiard, and so on. For instance, holes of billiards are related to negative curvature regions of the associated manifold.

It is easy to see this in general. Consider a billiard in arbitrary dimension $x_1, ..., x_N$ defined by some boundaries. Each point inside the allowed region $\mathbf{x}$ is characterized by the minimal distance to the boundary $M(\mathbf{x})$. We may "inflate" this into a manifold by adding an extra coordinate $z$, and defining the manifold embedded in the $N+1$-dimensional space by:

$$z^2 = k^2 [M(\mathbf{x})]^2 . \tag{6}$$

This has a symmetry $z \to -z$, so that quantum mechanically, wavefunctions may be classified according to their parity under this transformation. We recover the billiard as $k \to 0$, the appropriate boundary conditions are obtained by restricting to the subspace of odd functions in $z$.

As an example, we may promote a system of hard spheres of coordinate $r_i$ and radius $R$ to a manifold as:

$$z^2 = k^2 \left[ \min_{ij} |\mathbf{r}_i - \mathbf{r}_j| - R \right]^2 . \tag{7}$$

Therefore, studying classical or quantum dynamics on curved manifolds includes billiards as a limit.

## 4 Dynamics on curved manifolds and scaling of the quantum bounds

A Riemannian manifold is parametrized by a set of coordinates $(x_1, x_2, \ldots, x_N)$ and is defined by the metric

$$ds^2 = g_{ik}(\vec{x}) \, dx_i dx_k , \tag{8}$$

where the metric tensor is $g_{ik}(\vec{x}) = g_{ik}(x_1, x_2, \ldots, x_N)$ in $N$ dimensions, $g^{ik}$ its inverse and $g \equiv \det g_{ik}$ its determinant. It will be convenient from now onward to adimensionalize the coordinates by dividing by some *characteristic length $R$* of the system, for example a typical radius. The *classical Hamiltonian* for free motion in a manifold is

$$H_{cl} = \frac{1}{2\mu R^2} \, p_i g^{ik} p_k , \tag{9}$$

and the Lagrangian is thus

$$L = \frac{\mu R^2}{2} \, g_{ik} \, \dot{x}_i \dot{x}_k . \tag{10}$$

Here, we have made explicit the length scale $R$ and now $x_i$ and $g_{ij}$ are dimensionless variables. Other adimensional parameters are denoted $\vec{\alpha}$. The action and the length are respectively

$$\text{Action} = \int dt \, L(x, \dot{x}) \, , \quad \text{Length} = \frac{1}{\mu^{1/2}} \int dt \, \sqrt{L(x, \dot{x})} \, . \tag{11}$$

The extremization of both equations yields the same Euler-Lagrange equations, because $L$ itself is a constant of motion, so we conclude that classical trajectories are *geodesics*, covered at speed $\sqrt{2L/\mu} = \sqrt{2E/\mu}$.

There also exists a well-defined "free" *quantum dynamics* in curved geometry,

$$\hat{H}\Psi = -\frac{\hbar^2}{2\mu R^2}\nabla^2\Psi = i\hbar\frac{d\Psi}{dt} \, , \tag{12}$$

where the Hamiltonian is given by the invariant Laplacian on curved manifolds, i.e. the Laplace-Beltrami operator [11]

$$\nabla^2 = \frac{1}{\sqrt{g}}\frac{\partial}{\partial x_i}\sqrt{g}\,g^{ik}\frac{\partial}{\partial x_k} \, . \tag{13}$$

As a pure consequence of the rescaling in Eq.(12), the Heisenberg evolution of a generic can be re-written as

$$\hat{B}(t) = e^{i\hat{H}t/\hbar}\,\hat{B}\,e^{-i\hat{H}t/\hbar} = e^{-i\frac{1}{4\pi}\left(\frac{\ell_{th}}{R}\right)^2\nabla^2\frac{t}{\tau_{\text{Pl}}}}\,\hat{B}\,e^{i\frac{1}{4\pi}\left(\frac{\ell_{th}}{R}\right)^2\nabla^2\frac{t}{\tau_{\text{Pl}}}} \, , \tag{14}$$

where we have used the definition of the thermal de Broglie length (3) and divided and multiplied by $\tau_{\text{Pl}} = 1/\beta\hbar$. This implies that the adimensional observables evaluated at the *thermal time* $t_{\text{th}} = t/\tau_{\text{Pl}}$ are only a function of $R/\ell_{\text{th}}$. Furthermore, the characteristic time-scale of the problem is $\tau_{\text{Pl}}$. In what follows, we show that as a result of Eq.(14), the adimensional transport coefficients $\eta$ and Lyapunov exponent $\lambda$ are only a function of $R/\ell_{\text{th}}$ and hence obey Eq.(4).

## 4.1 Transport Coefficients

A nice way to introduce transport coefficients is with the Helfand-moment formalism, which allows one to write transport coefficients as diffusion's rates, see e.g. Refs. [17, 18]. Any transport coefficient $\alpha$ can be associated to a current $J_\alpha(x, p)$, a fluctuating quantity of order $\sqrt{V}$ (with $V \sim R^d$ the volume). The Green-Kubo formulas read [19]

$$\alpha_c = \frac{1}{VT}\int_0^\infty dt \, \langle J_\alpha(t)J_\alpha(0)\rangle_T \qquad \text{classical}, \tag{15a}$$

$$\alpha = \frac{1}{2VT}\int_0^\infty dt \, \text{Tr}\left\{[\hat{J}_\alpha(t), \hat{J}_\alpha(0)]_+ \frac{e^{-\beta H}}{Z}\right\} \quad \text{quantum}, \tag{15b}$$

where $\langle\cdot\rangle_T$ is the classical thermal average at temperature $T$ and $[\hat{A}, \hat{B}]_+ = \hat{A}\hat{B} + \hat{B}\hat{A}$. Introducing in both cases the Helfand function $G^\alpha(t)$ defined by $J_\alpha(t) = \sqrt{VT}\frac{d}{dt}G^\alpha(t)$ [17], the coefficients $\alpha$ may be written as

$$\alpha_c = \lim_{t\to\infty}\lim_{V\to\infty}\frac{\langle(G_\alpha(t) - G_\alpha(0))^2\rangle_T}{2t} \, , \tag{16a}$$

$$\alpha = \lim_{t\to\infty}\lim_{V\to\infty}\frac{\text{Tr}\left\{(\hat{G}_\alpha(t) - \hat{G}_\alpha(0))^2 e^{-\beta\hat{H}}\right\}}{2t} \, . \tag{16b}$$

We see this as diffusion of – or in the direction of – the $G$. The limit of infinite volume is taken first because we do not want the $G$ to saturate upon equilibration [3]. All in all, a robust definition of "transport" for any Riemannian manifold is to consider a multiple-connected surface and count the diffusion around some handles of an appropriate Helfand function.

Let us consider the case of the classical viscosity $\eta$, i.e.

$$G_\eta(t) = \sqrt{\frac{1}{R^d T}} \ \frac{1}{N} \sum_i^N q_i^x p_i^y \ . \tag{17}$$

Adimensionalizing momenta and coordinates, one has $G_\eta = \sqrt{\frac{\mu}{R^{d-2}}} \tilde{G}_\eta$, where $\tilde{G}_\eta$ is a adimensional quantity of purely geometrical content: it depends on the geodesic structure of the manifold. This means that in terms of the adimensional time $\tilde{t} = t/\sqrt{R^2 \mu/T}$, the dynamics only depend on the parameters through a time rescaling. Substituting $\tilde{t}$ in Eq.(16a), we get

$$\eta_c = \frac{\sqrt{T\mu}}{R^{d-1}} \ \lim_{\tilde{t} \to \infty} \lim_{V \to \infty} \underbrace{\frac{\langle (\tilde{G}_\alpha(\tilde{t}) - \tilde{G}_\alpha(0))^2 \rangle_T}{2\tilde{t}}}_{geometric} \ . \tag{18}$$

This is a purely classical expression. Dividing both sides by $\hbar$ we promote it to semiclassical:

$$\frac{R^d \eta_c}{\hbar} = \frac{R}{\ell_{\text{th}}} \ \lim_{\tilde{t} \to \infty} \lim_{V \to \infty} \underbrace{\frac{\langle (\tilde{G}_\alpha(\tilde{t}) - \tilde{G}_\alpha(0))^2 \rangle_T}{2\tilde{t}}}_{geometric} \ , \tag{19}$$

and we thus obtain the classical scaling of Eq.(5). Quantum mechanically, any transport coefficient reads

$$\alpha = \lim_{t \to \infty} \lim_{V \to \infty} \frac{\text{Tr}\left\{ (\hat{G}_\alpha(t) - \hat{G}_\alpha(0))^2 e^{\frac{1}{4\pi}\left(\frac{\ell_{\text{th}}}{R}\right)^2 \nabla^2} \right\}}{2t} \ , \tag{20}$$

where we have substituted the quantum Hamiltonian (12) $\hat{H} = -\frac{\hbar^2}{2\mu R^2} \nabla^2 = -\frac{1}{4\pi\beta}\left(\frac{\ell_{\text{th}}}{R}\right)^2 \nabla^2$ and the definition of $\ell_{\text{th}}$ (3). This expression, together with rescaling of Heisenberg observables in Eq.(14), shows that the quantum Helfland moments evaluated at $t_{\text{th}} = t/\tau_{\text{Pl}}$ are only a function of $R/\ell_{\text{th}}$ and other dimensionless quantities.

Consider now the quantum viscosity via Eq.(17). Using $p_j = -i\hbar \partial/\partial y_i$, one immediately gets

$$\hat{G}_\eta(t) = -i \frac{\hbar}{\sqrt{R^d T}} \ \frac{1}{N} \sum_{i=1}^N x_i \frac{\partial}{\partial y_i} = \frac{\hbar}{\sqrt{R^d T}} \tilde{G}_\eta \ , \tag{21}$$

and correspondingly,

$$\frac{R^d \eta}{\hbar} = \lim_{t_{\text{th}} \to \infty} \lim_{V \to \infty} \frac{\text{Tr}\left\{ (\tilde{G}_\alpha(t_{\text{th}}) - \tilde{G}_\alpha(0))^2 e^{\frac{\ell_{\text{th}}^2}{2R^2} \nabla^2} \right\}}{2t_{\text{th}}} = \Omega_\eta\left(\frac{R}{\ell_{\text{th}}}, \vec{\alpha}\right) \ , \tag{22}$$

where we have defined $\Omega_\eta$ a generic adimensional function, and $\vec{\alpha}$ the other dimensionless parameters appearing in $\nabla^2$. For large $R \gg \ell_{\text{th}}$, the linear term in $\Omega_\eta$ dominates, and $\hbar$ drops off, so we retrieve the classical result $\eta_c \propto \frac{\sqrt{\mu T}}{R^{d-1}}$ in Eq.(18), see also Eq.(5). The latter vanishes at zero temperature, hence displaying a clear violation of the bound [20]. Therefore, quantum effects must arise at length-scales $\ell_{\text{th}} \sim R$, as depicted in Fig.1a.

---

[3]However, when we compute transport with the Green-Kubo formula, there is always a direction in which we may choose periodic boundary conditions. If we keep track of the number of turns around these, we may omit the large $V$ limit.

## 4.2 Lyapunov exponent

Let us now turn to the Lyapunov exponent. In the classical case, trajectories are geodesics that the system runs through at velocity $v = \sqrt{NT/\mu}$ [4]. On manifolds, chaos has purely geometric origin and nearby chaotic geodesics separate exponentially with distance $\ell$ along them [10] as

$$\Delta(\ell) \sim \Delta_0 e^{\ell/s} \,, \tag{23}$$

where $s$ is the average path after which errors grow by $e$, denominated the "geodesic separation" or "characteristic path length" [10]. The classical Lyapunov exponent $\lambda_c$, which measures separation per unit time, scales with the velocity: $\lambda_c = \frac{v}{s}$. As discussed in Ref. [9], one has $s = \sqrt{N}R$ and therefore

$$\lambda_c = \sqrt{\frac{T}{\mu} \frac{1}{R}} = \frac{T}{\hbar} \frac{\ell_{\text{th}}}{R} \frac{1}{\sqrt{2\pi}} \,, \tag{24}$$

the classical scaling of Eq.(5). Thus, the classical Lyapunov exponent should scale with the square root of the temperature [9]. As mentioned above, this scaling with temperature has been found numerically in a classical spin-liquid of interacting spins on the kagome lattice [15].

In quantum systems, chaotic properties can be characterised by the quantum Lyapunov exponent, based on the square square-commutator [5]

$$C(t) = -\text{Tr}\left\{ [\hat{B}(t), \hat{A}(0)]^2 \frac{e^{-\beta \hat{H}}}{Z} \right\} \,, \tag{25}$$

where $\hat{B}(t)$ and $\hat{A}(0)$ are quantum observables in the Heisenberg representation at different times. In the case of underlying classical chaotic dynamics, the square commutator is expected to grow exponentially in time

$$C(t) \sim \varepsilon^2 e^{2\lambda t} \,, \tag{26}$$

where $\varepsilon$ is a small parameter that enables a separation of time scales between the early time expansion and large times saturation [6]. By studying the closely related regulated out-of-time-ordered-correlator (OTOC)

$$F(t) = \frac{1}{Z}\text{Tr}\left[ e^{-\frac{\beta}{4}\hat{H}}\hat{B}(t)e^{-\frac{\beta}{4}\hat{H}}\hat{A}e^{-\frac{\beta}{4}\hat{H}}\hat{B}(t)e^{-\frac{\beta}{4}\hat{H}}\hat{A} \right] \,, \tag{27}$$

the authors of Ref. [6] have shown that in quantum systems the rate of growth of chaos defined from above is bounded at low temperature by Eq.(2).

It is now well established that exponential time-dependence occurs only in presence of well-defined classical chaos $\hbar \to 0$ [21], or when some other parameter other than $\hbar$ , such as $\frac{1}{N}$ or $\frac{1}{d}$ (with $N$ the number of individual constituents with all-to-all interactions and $d$ the spatial dimension) goes to zero [7, 22] [5]. In this sense, the exponential growth of the commutator in quantum systems is an exception rather than a rule [24].

We now show that the adimensional Lyapunov, namely $\lambda\hbar/T$, is only a function of the ratio between the smallest geometric length-scale $R$ and the thermal de Broglie length $\ell_{\text{th}}$ [cf. Eq.(3)] also in the quantum case. Using the rescaling of generic observables in Eq.(14), the four point-correlators appearing in the regulated OTOC in Eq.(27) can be re-written as

$$F(t) = \text{Tr}\left\{ \left[ e^{\frac{1}{4\pi}\left(\frac{\ell_{\text{th}}}{R}\right)^2\left[\frac{1}{4}-it_{\text{th}}\right]\nabla^2}\hat{B}\, e^{\frac{1}{4\pi}\left(\frac{\ell_{\text{th}}}{R}\right)^2\left[\frac{1}{4}+it_{\text{th}}\right]\nabla^2}\hat{A} \right]^2 \right\} \,, \tag{28a}$$

---

[4]Classical geodesics are characterized by velocity $\sqrt{2E/\mu}$. At equilibrium, energy is evaluated via the equipartition theorem, i.e. $E = NT/2$, see Ref. [9].

[5]The same is true for the quantum Kolmogorov-Sinai entropy, see Ref. [23]

thus one immediately concludes that

$$\frac{F(t_{\text{th}})}{F(0)} = f_\lambda \left( \frac{R}{\ell_{\text{th}}}, \vec{\alpha}, t_{\text{th}} \right) , \tag{29}$$

with $f_\lambda$ a generic function. The same holds for the square-commutator $C(t)$ of Eq.(25). Hence it follows that, whenever the square-commutator grows exponentially i.e.,

$$C(t) \sim e^{2\lambda t} = e^{2\hbar\lambda/T \, t_{\text{th}}} , \tag{30}$$

the adimensional growth rate $\hbar\lambda/T$ must be a function only of the ratio $\ell_{\text{th}}/R$, i.e.

$$\frac{\lambda\hbar}{T} = \Omega_\lambda \left( \frac{R}{\ell_{\text{th}}}, \vec{\alpha} \right) . \tag{31}$$

For the limit $R \gg \ell_{\text{th}}$, the leading order of $\Omega_\lambda$ is $\propto \frac{\hbar}{\sqrt{\mu T}}$. Then $\hbar$ drops off and we recover the classical result in Eq.(24), see also Eq.(4). The latter shows that in the classical limit $\hbar\lambda_c/T \propto \ell_{\text{th}}/R \propto \hbar^2/\sqrt{T}$ violating the bound (2) at low temperatures. Therefore, quantum effects must arise at length-scales $\ell_{\text{th}} \sim R$ for which $\hbar\lambda_c/T = \mathcal{O}(1)$. As we discuss in the next section, these reconcile with the quantum bound to chaos, as depicted in Fig.1.

The Lyapunov exponent is in a sense different from the transport coefficients: the latter are well-defined unless the coefficient itself becomes infinite because the scaling of transport with the size is anomalous. In the case of the Lyapunov exponent, the very definition may lose sense because there is no regime in which the growth of the instability is exponential. Consider the Ehrenfest time at which an initial packet has spread throughout the phase-space volume. It may be estimated as

$$\varepsilon^2 e^{2\lambda T_{\text{Ehr}}} \sim L , \tag{32}$$

where $L$ is a diameter of phase-space. The existence of a Lyapunov regime requires that there is a small parameter $\epsilon \sim \hbar$ (or $\epsilon \sim 1/N$) such that the Ehrenfest time

$$T_{\text{Ehr}} \sim \frac{1}{2\lambda} \ln \frac{L}{\epsilon} \tag{33}$$

is sufficiently large. There are cases when there is no Lyapunov regime at all. This is what is expected of spin chains with local interactions and finite spin representations [24]. As a consequence, quantum effects act on a Lyapunov exponent, not necessarily limiting its value, but they may also make their definition inapplicable.

## 5 Quantum mechanisms for the bound to chaos on a simple model

The goal of this section is to understand how quantum mechanics intervene to enforce the bound to chaos (2) at low energies. To do this, we start by exploring the simplest models describing a free particle moving on two-dimensional curved surfaces. In this context, it is more appropriate to reason in terms of energy $E$ rather than of temperature

$$T \quad \longrightarrow \quad E . \tag{34}$$

At equilibrium, the average energy per degree of freedom is related to temperature by the standard equipartition theorem. In a finite system, energy is a fluctuating quantity, and it seems more natural, rather than the thermal length $\ell_{\text{th}}$ [cf. Eq.(3)], to consider the *de Broglie wavelength*

$$\ell_{\text{dB}} = \frac{2\pi\hbar}{p} = \frac{2\pi\hbar}{\sqrt{2\mu E}} , \tag{35}$$

where $p$ is the momentum per degree of freedom.

We identify three quantum mechanisms responsible for the bounds. (1) Size: the classical description breaks down when $\ell_{dB}$ is of the order of the system's size. This is the most trivial scenario in which quantum fluctuations extend over all the system and do not allow any semi-classical description. The typical example is one of the spin-$S$ models with small $S$. (2) Avoidance of curved regions. (3) Spreading of wave-packets in competition with the curvature. In what follows, we first introduce the simple model given by a free particle on a curved surface, constructed by joining a surface of constant negative curvature to a cylinder, as in Fig.2. Then, we illustrate the effects (2) and (3) on such a toy model.

## 5.1 The free geometry

The metric reads

$$ds^2 = g_{\mu\nu}dx^\mu dx^\nu = \begin{cases} R^2(d\tau^2 + \sinh^2\tau\, d\phi^2) & \text{for} \quad 0 \le \tau < \tau_x, \\ R^2(d\tau^2 + \underbrace{\sinh^2\tau_x}_{\text{const.}}\, d\phi^2) & \text{for} \quad \tau_x < \tau < \tau_x + \tau_L, \end{cases} \tag{36}$$

where $R\tau$ is the radial geodesic distance and $\phi$ is the angle. For $0 \le R\tau < R\tau_x$, Eq.(36) corresponds to a representation in geodesic polar coordinates of the surface with constant negative Gaussian curvature $K = -1/R^2$, also known as the *pseudosphere*. At $\tau = \tau_x$, the pseudosphere is matched with a cylinder, that has a reflecting wall at $\tau = \tau_L$.

### 5.1.1 The pseudosphere

The pseudosphere constitutes the paradigmatic model for chaotic dynamics on manifolds. The classical and quantum properties of such metric have been discussed at length in Ref. [11], to which we refer for all the details. Surfaces of constant negative curvature can not be globally embedded in a three-dimensional Euclidean space [25]. Hence, a possible way to visualize this surface is to consider an embedding in the Minkowskian space. There, the psudosphere appears as one sheet of a two-sheeted hyperboloid, as in Fig.2. A first obvious reparametrization in terms of pseudospherical polar coordinates $y_1 = R\sinh\tau\cos\phi$, $y_2 = R\sinh\tau\sin\phi$ and $y_0 = R\cosh\tau$ (satisfying the condition $y_1^2 + y_2^2 - y_0^2 = -R^2$) leads to the metric (36) for $0 \le \tau < \tau_x$. The invariant volume element is $dV_{\text{Hyp}} = R^2\sinh\tau d\tau d\phi$. From the polar coordinates, one can readily deduce a different parametrization: the hyperbolic plane or *Poincaré disk*. Points on the Poincaré disk are obtained via a stereographic projection onto the $y_1, y_2$ plane about the point $(0,0,-R)$. In Fig.2, the hyperbolic plane is represented pictorially by Escher's Circle limit IV[6]. A point of coordinates $(R\tau, \phi)$ in the hyperboloid is described by polar coordinates $(Rr, \phi)$ on the disk of radius $R$ with $r = \tanh\frac{\tau}{2}$. With these coordinates, the invariant volume element reads $dV_{\text{Hyp}} = 4R^2 r\, dr d\phi/(1 - r^2)^2$. The boundary of the disc $(r = 1)$ corresponds to points at infinity of the hyperboloid $(\tau = \infty)$. Therefore, by considering a finite portion of pseudosphere for $\tau \le \tau_x$, we limit the Poincaré disk up to $r \le \tanh\tau_x/2 \equiv x$. The volume of the curved (chaotic) geometry is therefore

$$\text{Vol}_{\text{Hyp}} = \int dV_{\text{Hyp}} = 4\pi R^2 \sinh^2\frac{\tau_x}{2}. \tag{37}$$

### 5.1.2 The cylinder "lead"

At $\tau = \tau_x$, the hyperboloid is matched with a cylinder of radius $R\sinh\tau_x$ and height $L = R\tau_L$, whose metric is given by Eq.(36) for $\tau_x < \tau < \tau_x + \tau_L$. The invariant volume element

---

[6]Escher's *Circle limit I-IV* series represent tesselations of the hyperbolic plane. They illustrate discrete subgroups acting on the unit disc. See Ref. [26].

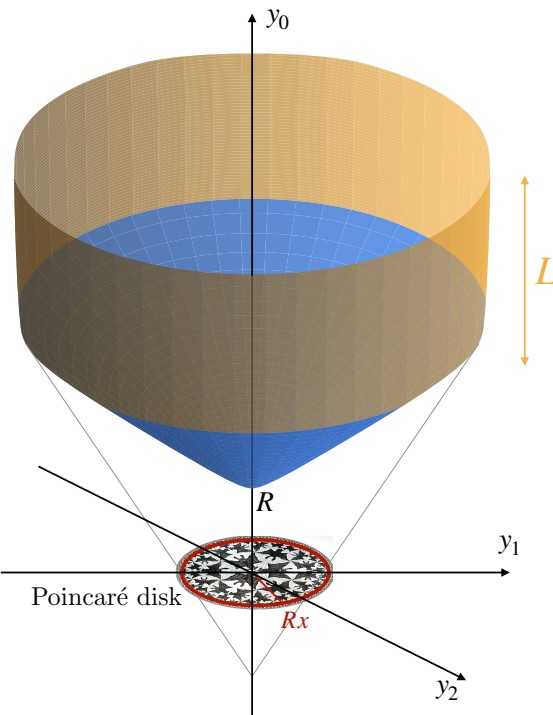

Figure 2: Geometry of the surface with zero and constant negative curvature. The three-dimensional plot represents a section of the hyperboloid with constant negative curvature $-1/R^2$ (blue) and the cylinder of length $L = \tau_L R$ (orange) in the Minkowski metric (see discussion in Sec.5). Points of the hyperboloid are projected onto the $(y_1, y_2)$ plane with lines originating in $-R$. The points lie in the interior of the Poincaré disk of radius $R$. The latter is pictorially represented by Escher's Circle Limit IV. The presence of the cylinder starting from $\tau_x$ corresponds to limiting the radius of the Poincaré disk to $Rx$ with $x < 1$.

$dV_{\text{Cil}} = R^2 \sinh \tau_x \, d\tau \, d\phi$ allows to compute the cylinder volume

$$\text{Vol}_{\text{Cyl}} = \int dV_{\text{Cyl}} = 2\pi R^2 \tau_L \sinh \tau_x \,. \tag{38}$$

Since this surface has zero Gaussian curvature $K = 0$, the free motion on a cylinder is completely regular. Its role is to emphasise the effect of negative curvature to the bound to chaos, by comparing the wavefunctions in the two regions.

Summarizing, the relevant length-scale of this geometry is the radius of the pseudosphere $R$, while the dimensionless parameters are $\tau_x$ and $\tau_L$, that represent the length of the pseudosphere and the height cylinder respectively. Through these, we can tune the ratio between the volume of the chaotic region and the total one, i.e.

$$\frac{\text{Vol}_{\text{Hyp}}}{\text{Vol}_{\text{Hyp}} + \text{Vol}_{\text{Cyl}}} = \frac{\tanh(\tau_x/2)}{\tanh(\tau_x/2) + \tau_L} = \frac{x}{x + \tau_L} \,. \tag{39}$$

We choose large dimensionless parameters $\tau_x \gg 1$ and $\tau_L \gg 1$, such that the volume of the model is always very large. As a result, we immediately overcome the first quantum mechanism due to the limitations of the system size that we get if the circumference of the cylinder is small with respect to the de Broglie length, as well as avoiding a small Ehrenfest time. This will allow us to concentrate on the effect of the radius of curvature $R$.

## 5.2 Classical dynamics of the free model

The associated classical Hamiltonian is

$$H_0 = \frac{1}{2\mu R^2} \begin{cases} p_\tau^2 + \dfrac{I_\phi^2}{\sinh^2 \tau} & \text{for} \quad 0 \le \tau < \tau_x, \\[2ex] p_\tau^2 + \underbrace{\dfrac{I_\phi^2}{\sinh^2 \tau_x}}_{\text{const.}} & \text{for} \quad \tau_x < \tau < \tau_x + \tau_L, \end{cases} \tag{40}$$

where $I_\phi$ is the angular momentum. For $0 \le \tau < \tau_x$, the orbits correspond to geodesics on the pseudosphere, these can be visualized on the Poincaré disk as diameters of circles orthogonal to the boundary [11], see Fig.3. This region is chaotic, in the sense that nearby geodesic separate exponentially fast in time with the classical Lyapunov exponent $\lambda_c$ given by Eq.(41). On the other hand, for $\tau_x < \tau < \tau_x + \tau_L$ the system undergoes regular free motion on the cylinder. Inside the pseudosphere, the presence of the angular momentum generates the standard classical repulsive potential at small distances $\propto I_\phi^2/\sinh^2 \tau$. For small values of $I_\phi$ the particle enters in the pseudosphere also at small $\tau$, while for large values it explores only its boundary and spends most of the time in the cylinder. Note that, once integrability is broken (see below), this corresponds to an ergodic equilibrium distribution that is uniform with respect to the metric, see Fig. 9 in the Appendix.

The classical Lyapunov exponent of the pseudosphere has the scaling of Eq.(24) and reads

$$\lambda^c = \sqrt{\frac{2E}{\mu R^2}} \,, \tag{41}$$

where $R$ is the radius of the pseudosphere that here plays the role of the geodesic separation $s$ for one degree of freedom [cf. Eq.(23)]. Since all the chaotic contribution comes from the curved region, the total Lyapunov exponent is given by the the fraction of time the particle spends in the curved chaotic region times its $\lambda^c$ in Eq.(41), as

$$\lambda_{tot}^c = \lambda^c \frac{\text{Vol}_{\text{Hyp}}}{\text{Vol}_{\text{Hyp}} + \text{Vol}_{\text{Cyl}}} \,, \tag{42}$$

where, because of ergodicity, the ratio of times is given by the ratio of volumes that can be computed explicitly from the metric [cf. Eq.(39)].

## 5.3 Small perturbation and chaotic behaviour

The model given by $H_0$ is not our end product. In fact the model at this level is integrable due to axial rotational invariance ($H_0$ has two constants of motion: angular momentum and energy). What became of the chaoticity inside the hyperboloid? The answer is simple: two neighbouring trajectories starting in the cylinder separate exponentially when they enter the pseudosphere until they start returning towards the cylinder and they approach one another exponentially. As such, motion on this surface is exponentially unstable and it is enough to add some small arbitrary perturbation to break the angular momentum conservation. Therefore, any small integrability-breaking term added to the Hamiltonian (40) shall make the model fully chaotic with a Lyapunov exponent independent from the perturbation. We consider the following slightly perturbed Hamiltonian

$$H = H_0 + \gamma V(\tau, \phi) \quad \text{with} \quad \gamma \ll 1 \,, \tag{43}$$

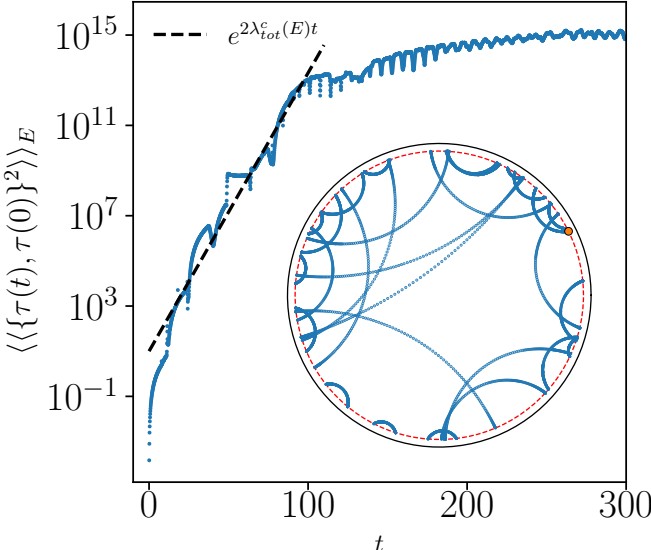

Figure 3: Chaotic dynamics of the free particle on the surface with zero and constant curvature. Main plot: square of the classical Poisson brackets (45) averaged over 50 initial conditions distributed according to the equilibrium distribution. The dashed black line represents the exponential growth with rate given by twice the total classical Lyapunov exponent (42). Inset: single trajectory on the Poincaré disk with initial angular momentum $I_0 = 2.3$. Parameters of the simulation: $\tau_x = 3.7$, $\tau_L = 20$, $E = 3.5$, $R = 1$ and perturbation with $p = (2, 5, 7, 10)$ with $\gamma_p = (1, 2, 0.5, -2)$.

where $H_0$ is given by Eq.(40) and $V(\tau, \phi)$ is a perturbation, whose specific form does not change the results. In particular, for a reasons that will be more transparent in the quantum case, we chose $V(\tau, \phi)$ as a sum over some integers $p$ of the following potential

$$V_p(\tau, \phi) = \cos(p\phi) \begin{cases} \left( \frac{\tanh \tau/2}{x} \right)^p & \text{for} \quad \tau < \tau_x, \\ 1 & \text{for} \quad \tau_x < \tau < \tau_x + \tau_L, \end{cases} \tag{44}$$

where $\cos(p\phi)$ breaks the conservation of the angular momentum. *At each step we check that the results are independent of the form of the perturbation, and are given by the chaotic parameters of the hyperboloid itself.* This is exemplified by the inset of Fig.3 where we show a single trajectory on the Poincaré disk, that appears like the pure unperturbed one of a chaotic curved billiard. We also test the equilibration, verifying that the long-time average of observables corresponds to the equilibrium distribution, see e.g. Fig.4 in App. A. See the same Appendix for all the details on the numerical implementation.

Secondly, we explore the classical Lyapunov exponent of the system. Since all the chaotic contribution comes from the pseudosphere, the total Lyapunov exponent is given by Eq.(42). To work in analogy to the quantum problem, we probe $\lambda_{tot}^c$ by looking at the square of the Poisson brackets, equivalent to the square-commutator of Eq.(25). We study

$$c_{cl}(t) = \langle\!\langle \{\tau(t), \tau(0)\}^2 \rangle\!\rangle_E = \left\langle\!\!\left\langle \left( \frac{d\tau(t)}{dp_\tau(0)} \right)^2 \right\rangle\!\!\right\rangle_E, \tag{45}$$

where $\langle\!\langle \cdot \rangle\!\rangle_E$ represents the average over different initial conditions sampled according to the equilibrium distribution at energy $E$. Being proportional to the derivative of the trajectory to respect to the initial conditions, $c_{cl}(t)$ grows exponentially fast in time with a rate given by

twice $\lambda_{tot}^c$, see Fig.3. We notice that, in complete analogy with the quantum case, this quantity yields the *annealed* Lyapunov exponent that is in general dominated by rare events.

### 5.4 Quantum dynamics

In the quantum problem, the free evolution of a particle on a manifold is determined by the Schrödinger equation where the Laplace-Beltrami operator (13) acts as the Hamiltonian

$$\hat{H}_0 \Psi = -\frac{\hbar^2}{2\mu R^2} \nabla^2 \Psi = E\Psi \ . \tag{46}$$

The normalization condition for the wave functions $\Psi$ is given by $1 = \int dV \, |\Psi(\tau,\phi)|^2 = \int \sqrt{g} \, |\Psi(\tau,\phi)|^2 \, d\tau d\phi$ , where $g = \det g_{ij}$ is the determinant of the metric. As a matter of convenience, we consider the transformed $\Psi(\tau,\phi)$ into $\Phi(\tau,\phi)$ by the following relation

$$\Phi(\tau,\phi) = g^{1/4} \Psi(\tau,\phi) \ , \tag{47}$$

such that normalization condition then becomes $1 = \int |\Phi|^2 \, d\tau d\phi$. The Laplace-Beltrami operator $\nabla^2$ (13) associated to the metric (36) reads

$$\nabla^2 = \begin{cases} \dfrac{1}{\sinh\tau} \dfrac{\partial}{\partial\tau} \left( \sinh\tau \dfrac{\partial}{\partial\tau} \right) + \dfrac{1}{\sinh^2\tau} \dfrac{\partial^2}{\partial\phi^2} & \text{for} \quad \tau < \tau_x \,, \\[4mm] \dfrac{\partial^2}{\partial\tau^2} + \underbrace{\dfrac{1}{\sinh^2\tau_x}}_{\text{const.}} \dfrac{\partial^2}{\partial\phi^2} & \text{for} \quad \tau_x < \tau < \tau_x + \tau_L \,. \end{cases} \tag{48}$$

We apply the Schrödinger equation to $\Psi(\tau,\phi) = g^{-1/4}\Phi(\tau,\phi)$ and obtain the following equation for $\Phi$:

$$\begin{cases} -\dfrac{\hbar^2}{2\mu R^2} \left( \dfrac{\partial^2}{\partial^2\tau} + \dfrac{1}{\sinh^2\tau} \dfrac{\partial^2}{\partial\phi^2} \right) \Phi + V_{\text{eff}}(g)\Phi \\[4mm] -\dfrac{\hbar^2}{2\mu R^2} \left( \dfrac{\partial^2}{\partial^2\tau} + \dfrac{1}{\sinh^2\tau_x} \dfrac{\partial^2}{\partial\phi^2} \right) \Phi \end{cases} = E\Phi \ , \tag{49}$$

where the effective potential reads

$$V_{\text{eff}}(g) = \frac{\hbar^2}{2\mu R^2} \frac{1}{4} \left( 1 - \frac{1}{\sinh\tau^2} \right) \ . \tag{50}$$

Therefore, the quantum effect of curvature is to generate a repulsive potential $\propto \hbar^2$ composed of a "centrifugal" term $\propto \frac{1}{\sinh^2\tau}$ and a constant energy step, i.e.,

$$\Delta = \lim_{\tau \to \infty} V_{\text{eff}}(g) = \frac{\hbar^2}{2\mu R^2}\delta \quad \text{with} \quad \delta = \frac{1}{4} \ . \tag{51}$$

Divergences as $\sim \frac{1}{\sinh\tau^2} \sim \tau^{-2}$ for $\tau \to 0$ are know to give an anomalous bound state in quantum mechanics [27]. Here however, we are interested in higher energy levels.

The eigenenergies and eigenfunctions of $H_0$ can be determined exactly via separation of variables $\Psi(\tau,\phi) = e^{im\phi}F(\tau)$, where $m$ is the integer eigenvalue of the angular momentum and $F(\tau)$ is determined by solving the radial Schrödinger in the pseudoshere and the cylinder

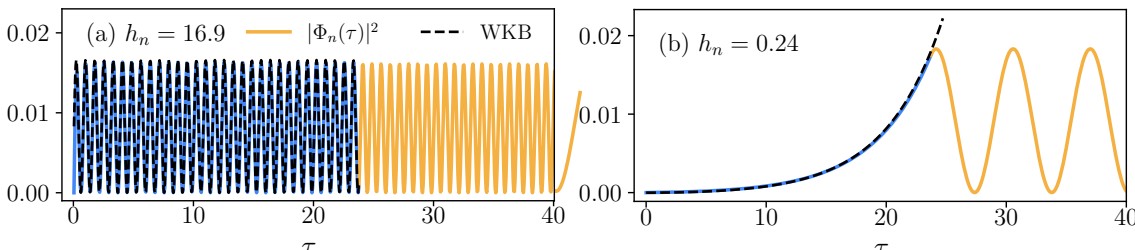

Figure 4: Effect of the energy gap $\Delta$ (51) on the exact wave-functions. Unperturbed eigenstates (52) with energy above the gap (a) and right below it (b). In the pseudosphere, we compare with the WKB predictions (dashed black) for the infinite cone (B.10). Parameters: $\tau_x = 24$ and $\tau_L = 100$ and zero angular momentum $m_n = 0$. In order to emphasise the repulsive effect of the pseudosphere at low energies, we plot only a portion of the $X$-axis up to $\tau = 40$, while the total range is $\tau_x + \tau_L = 124$.

separately and then by imposing continuity conditions in $\tau_x$. All the details are reported in the App.C. The resulting normalized *eigenstates* $\Phi_n = g^{1/4}\Psi_n$ read

$$\Phi_n(\tau, \phi) = \frac{e^{im_n\phi}}{\mathcal{N}_n} \tag{52}$$

$$\times \begin{cases} \sqrt{\sinh\tau}\, P_{m_n}^{\ell_n}(\cosh\tau) & \text{for} \quad \tau < \tau_x, \\ \sqrt{\sinh\tau_x}\, (C_n\cos[\kappa_n(\tau-\tau_x)] + D_n\sin[\kappa_n(\tau-\tau_x)]) & \text{for} \quad \tau_x < \tau < \tau_x + \tau_L, \end{cases}$$

where the coefficients read, $\forall \kappa_n \neq 0$,

$$C_n = P_m^{\ell_n}(\cosh\tau_x) \tag{53}$$

$$D_n = \frac{1}{\kappa_n}\left[P_{m+1}^{\ell_n}(\cosh\tau_x) + \left(m+\frac{1}{2}\right)\coth\tau_x\, P_m^{\ell_n}(\cosh\tau_x)\right].$$

The quantization condition is found by solving the following condition

$$\tan(\kappa_n\tau_L) = -\frac{C_n}{D_n}, \tag{54}$$

while the adimensional *eigenvalues* $h^{(0)}$ are related to the degree $\ell_n$ and the momentum $\kappa_n$ in (52) by

$$\ell_n = -\frac{1}{2} + \sqrt{\frac{1}{4} - h_n^{(0)}}, \quad h_n^{(0)} = \kappa_n^2 + \frac{m_n^2}{\sinh^2\tau_x}. \tag{55}$$

This allows exploring the effect of the curvature gap $\Delta$ on their behaviour. We plot the wave-functions $\Phi_n$ [cf. Eq.(52)] in Fig.4, contrasting their behavior at high energies (a) and for energies and right below the gap (b). To emphasise this effect, we consider the solutions with $m_n = 0$, that do not experience the classical repulsive potential in the pseudosphere $\propto m^2/\sinh^2\tau$. We compare it with the semi-classical WKB approximation, which holds for states with high quantum numbers $n \gg 1$ [cf. Eq.(B.10) in the App. B]. At high energy (Fig.4a), the system does not see the gap $\delta$ and the wave-function is completely delocalized across the cylinder and the pseudosphere. On the other hand, right below $\Delta$ (Fig.4b), although the wave function undergoes regular oscillations in the cylinder, the presence of the gap causes $\Phi_n$ to be an evanescent wave, exponentially damped in the region with constant negative curvature.

## 5.5 Mechanism 2) Avoidance of curved regions

We now argue how the presence of a repulsive potential $\Delta$ (51), arising from the curvature, enforces the bound at low energies. For this discussion, we do not need to consider the full Hamiltonian, and just the properties of $\hat{H}_0$ will suffice. The presence of an energy gap $\Delta$ naturally affects the Lyapunov exponent (41) and the bound to chaos. We assume a *semi-semi classical* approach, in which the oscillations in $\tau$ are short with respect to $\tau_x$. The state is in the semi-classical regime (large quantum numbers $n \gg 1$): the particle obeys the classical equations of motion, yet it may have energy as low as $O(\hbar^2)$. In the classical limit at high energies, the hyperbolic region provides the chaoticity to the system, with the Lyapunov exponent proportional to $\lambda_c$ in Eq.(41) [7]. However, when the energies are comparable to $\Delta$, the particle slows down inside the chaotic region $p \propto \sqrt{E - \Delta}$, leading to the following Lyapunov exponent

$$\lambda^c = \frac{p}{\mu R} = \sqrt{\frac{2}{\mu R_0^2}} \sqrt{E - \Delta} \tag{56}$$

and

$$\frac{\hbar \lambda^c}{E} = \sqrt{\frac{2\hbar^2}{\mu R^2}} \frac{\sqrt{E - \Delta}}{E} = \frac{\ell_{dB}}{\pi R} \sqrt{\frac{h - \delta}{h}} \,, \tag{57}$$

where $h$ ($\delta$) correspond to the dimensionless energy $E = \hbar^2 h / 2\mu R^2$ (gap $\Delta = \hbar^2 \delta / 2\mu R^2$) and we have substituted the definition of $\ell_{dB}$ from Eq.(35). For $E \gg \Delta$ we retrieve the classical dimenstionless Lyapunov exponent $\sim 1/\sqrt{E}$. Then, the function has a maximum for $E = 2\Delta$ and it decreases up to vanishing for $E = \Delta$, see Fig.6a. The maximum value of $\frac{\hbar \lambda}{E}$ – the greatest approach to the bound – happens at $\ell_{dB}(2\Delta) = 2\sqrt{2}\pi R$, i.e. where quantum and geometric lengths are comparable. We shall argue that this is a general fact.

We corroborate this semi-semi-classical picture with a *numerical investigation* of the perturbed model described by the quantum version of the Hamiltonian (43). The full Hamiltonian is written as a $N_{cut} \times N_{cut}$ matrix in the unperturbed basis, which is in turn fixed by solving numerically the condition (54) for $N_{cut} \gg 1$ number of states. The cutoff number of states $N_{cut}$ is then increased up to the convergence of the results. We refer to the App.C for all the details on the chaotic spectral properties with the perturbation.

We study the quantum Lyapunov exponent, by looking at the *microcanonical* version of the square-commutator in Eq.(25), see, e.g., Ref. [28], i.e. $c(t) = -\langle n_0 | [\hat{\tau}(t), \hat{\tau}(0)]^2 | n_0 \rangle$, where $|n_0\rangle$ in an eigenstate of the perturbed Hamiltonian $\hat{H}$, that we study in the classical limit $n_0 \gg 1$. In Fig. 5, we show the evolution of the square commutator for different eigenstates $n_0$. On the left, the results at high energies $h_{n_0} \gg 1$ display a $c(t)$ that grows exponentially with twice the classical total Lyapunov exponent [cf. Eq.(42)] at the corresponding energy. Due to quantum interference, this growth holds only in a time window – the Lyapunov regime – that ends at the Ehrenfest time $T_{Ehr} \propto \log h_{eff}^{-1} \propto \log n_0$. Indeed, the Lyapunov regime is longer for higher energies. On the right of Fig. 5, we consider a square-commutator obtained a $|n_0\rangle$ right below the gap $h_{n_0} \lesssim \delta$. We check that this eigenstate $|n_0\rangle$ is sufficiently chaotic (cf. App.C), however, the $c(t)$ does not display an exponential regime, as shown by the polynomial growth in the inset, in agreement with our picture for the bound to chaos.

---

[7] Note that this Lyapunov exponent shall be multiplied by the ratio of times the particle spends in the chaotic region over the total time. In the case of our toy model this is given by Eq.(39).

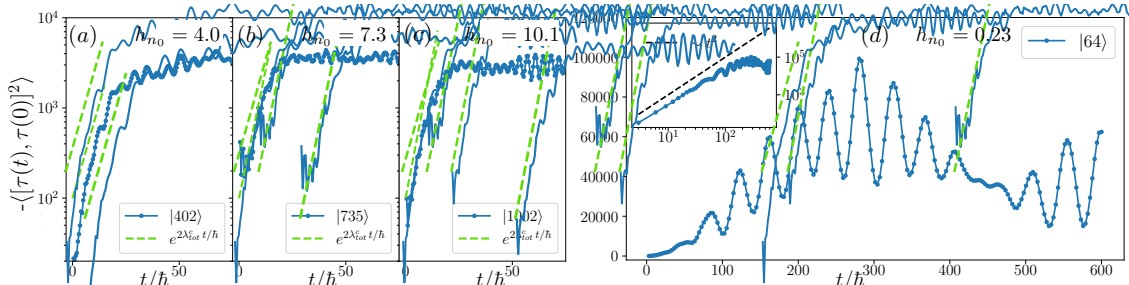

Figure 5: (a-c) Exponential growth of the square-commutator at high energies. Simulations with different eigenstates $|n_0\rangle = |402\rangle, |735\rangle$ and $|1002\rangle$ from left to right. (Dotted blue) numerical results converged upon increasing $N_{cut}$, here $N_{cut} = 1630$. (Dashed green) semiclassical prediction: the square commutator grows in time exponentially with the classical rate $\lambda_{tot}^c$ (42). Parameters: $\tau_L = 20$, $\tau_x = 3.7$ and perturbation $p = (10, 7, 5, 2)$ with $\gamma_p = (-0.8, 1.6, 2, -3)$, such that the relative error induced by the perturbation is $\langle \hat{V} \rangle / E = 0.12, 0.12, 0.02$ respectively. (d) Absence of exponential growth at low energies. Growth of the square-commutator for the eigenstate $|n_0\rangle = |64\rangle$. Numerical results converged upon increasing $N_{cut}$, here $N_{cut} = 800$. (Inset) Log-log plot of the same data displaying a polynomial growth. Parameters: $\tau_L = 40$, $\tau_x = 4$ and perturbation $p = (10, 7, 5, 2)$ with $\gamma_p = (-0.5, 1, 1.5, -2.5) \times 10^{-1}$ and relative error $\langle \hat{V} \rangle / E = 0.2$.

### 5.6 Mechanism 3) Spreading of the wave packet

We have seen that curvature, the cause of chaos, is also a source of repulsion for the particle at the quantum level. However, this cannot be the whole mechanism, since we are always free to apply a potential that favours visiting the most curved part. This situation is equivalent to studying what happens the quantum solution of a free particle on a pseudosphere with an hard wall at some $\tau_x$. See App.D for the classical and quantum solutions of the pseudosphere with a boundary. In this section we shall restrict to this example. This will allow us to identify a different mechanism.

We restrict ourselves to the pseudosphere and we compute the square-commutator initializing the system in a localized Gaussian wave-packet with variance $\sigma_0 = \frac{\hbar}{2\Delta p}$ determined by maximizing the uncertainty principle.

Classically, the spatial separation between two geodesics (with different angular initial conditions $\delta\phi(0)$) is computed via Eq.(36) and reads

$$\left(\frac{ds(t)}{ds(0)}\right)^2 = \sinh^2(\tau(t)/R)\left(\frac{d\phi(t)}{d\phi(0)}\right)^2 \sim \sinh^2(\tau(t)/R) \sim e^{2\lambda_c t} \quad \text{for} \quad t \to \infty\,, \quad (58)$$

with $\tau$ being now a dimensional variable. In the second line, we have used that angular variations remain almost constant in time, see App. D.1. All the exponential growth comes from the way we are computing distances, hence from the geometry, so it is very easy to analyze. In the quantum case, the corresponding square commutator is $c(t) = -\langle \sinh^2 \hat{\tau}(t) [\hat{\phi}(t), \hat{I}(0)]^2 \rangle$. By evaluating the expectation value over a factorized wave-packet in the angular and radial coordinates we can approximate $c(t) \simeq \langle \sinh^2(\hat{\tau}(t)/R) \rangle \sim \langle e^{2\hat{\tau}(t)/R} \rangle$. A straightforward calculation of the Gaussian integral in the expectation value leads to

$$c(t) \sim \langle e^{2\hat{\tau}(t)/R} \rangle = \exp\left[2\left(\frac{\ell_{dB}}{4\pi R}\frac{p}{\Delta p}\right)^2 + 2\lambda_c t + 2\left(\frac{\Delta p}{p}\right)^2 \lambda_c^2 t^2\right]. \quad (59)$$

The interpretation of this expression is clear: The centre of the wave-packet contributes with $c(t) \sim \exp(2\lambda_c t)$, while the front of the wave-packet dominates the Gaussian integral leading

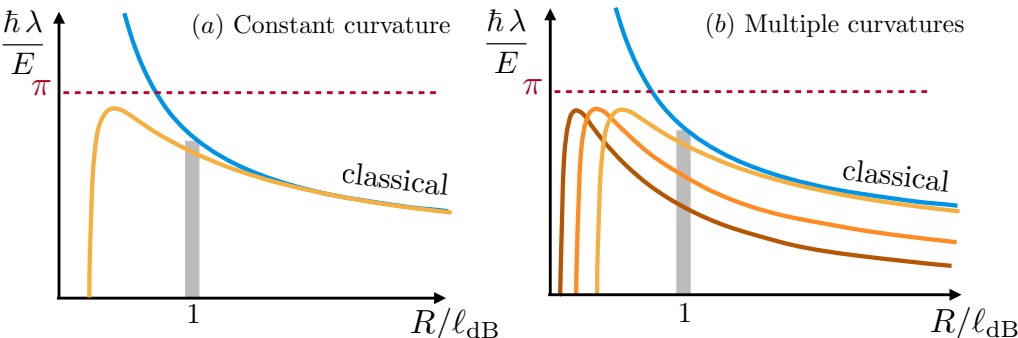

Figure 6: Pictorial representation of the adimensional Lyapunov exponent (57) as a function of the ratio between the curvature radius and the de Broglie length $\ell_{dB}$ (35). The classical limit (blue curve) diverges as $\sim E^{-1/2}$ for small energies $E$. The bound to chaos is implemented by the quantum effects of curvature for $E \sim 2\Delta$ (57), when $\ell_{dB} \sim R$. (a) Single curvature radius $R$. (b) Multiple curvature length-scales $R_i$. The bound holds at zero energy (temperature) in the presence of a hierarchy of diverging lengths.

to the third term in Eq.(59). Therefore, the classical Lyapunov regime dominates up to times $\lambda_c t \leq \lambda_c t_1 = (p/\Delta p)^2$, after which a superexponential growth $c(t) \sim \exp(2\lambda_c^2 t^2)$ kicks in. Interestingly, we notice that the same regime is well known to occur in the Loschmidt echo dynamics, see e.g. Ref. [29] and it has been recently discussed for the square-commutator in Ref. [30]. Hence, in order to appreciate the exponential growth of the square-commutator one needs

$$\lambda_c t_1 = \left(\frac{p}{\Delta p}\right)^2 \gg 1 \,. \tag{60}$$

This condition is equivalent to the requirement of an initial wave-packet well localized in momentum. At the same time, we also need to assure that the initial value $c(0)$ and $c(t_1)$ are parametrically different, that is

$$\frac{R}{\ell_{dB}} \gg \frac{1}{4\pi} \,. \tag{61}$$

In this simple scenario, the Lyapunov regime dominates whenever the conditions (60) and (61) are satisfied. While the former depends only on the structure of the initial wave-packet, the latter sets a condition between the de Broglie length and the radius of curvature of the model, exactly in the spirit of Sec. 4.2.

This quantum mechanism is paradoxical, in that it enhances separation of trajectories rather than suppressing it, but limits the Lyapunov regime defined as the one in which separation is simply exponential in time. This makes the definition of the Lyapunov exponent, and thus the bound in Eq.(2), ill defined.

# 6 Approach of the bound to chaos at the lowest temperature: many length scales

Up to now, we have discussed how the bound is implemented at small but finite energy, as in Fig.6a. One lesson we have learned is that the approach to the bound reaches its closest point at a temperature such that the de Broglie length is comparable to the length of the geometric features responsible for the chaos. In order to approach the bound in the limit of zero energy

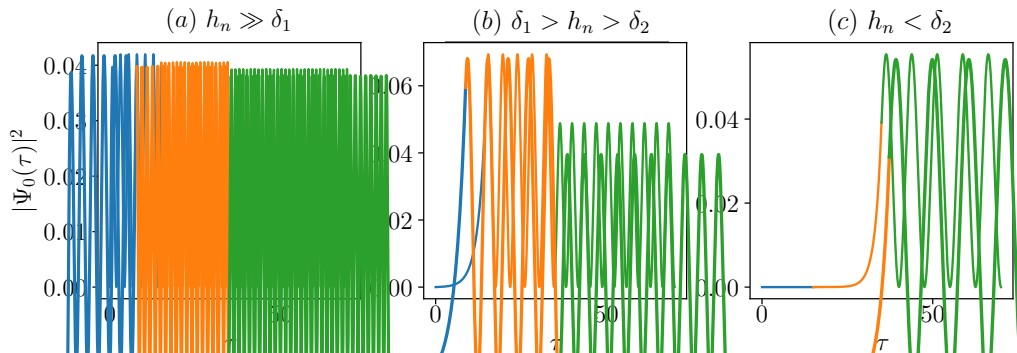

Figure 7: Effect of the multiple lengths on the toy model with multiple radius of curvature. Unperturbed eigenstates with $n = 2$ at zero angular momentum $m_n = 0$. (a) Energy much above the gaps $h_n^{(0)} = 7.8 \gg \delta_1, \delta_2$. (b) Energy smaller than the first gap $\delta_1 > h_n^{(0)} = 0.8 > \delta_2$. (c) Energy much smaller than the lowest gap $h_n^{(0)} = 0.2 \ll \delta_2$. Parameters: $\tau_1 = 15$, $\tau_2 = 35$ and $\tau_L = 35$, $R_2/R_1 = 2$.

(temperature), one then needs a hierarchy of length scales, since longer and longer ones will provide chaos at lower and lower temperatures.

As an example, let us consider a surface designed by of $N$-constant negative curvatures with radii $R_i$ in regions $\tau_i < \tau < \tau_{i+1}$, and for $\tau_n < \tau < \tau_n + \tau_L$ a small cylinder of length $L = R_n \tau_L$ and radius $R_n \sinh \tau_n$. We fix for definiteness $R = R_1 \leq R_2 \leq \cdots \leq R_n$. The largest instabilities are associated to regions with the smallest $R_i$, as

$$\lambda_i^c = \sqrt{\frac{2E}{\mu R_i^2}} \quad \text{with} \quad i = 1, \ldots, n\,, \tag{62}$$

that in turn are characterized by the largest energy gaps

$$\Delta_i = \frac{\hbar^2}{2\mu} \frac{1}{4R_i^2}\,. \tag{63}$$

Hence, regions with the smallest lengths scale $R_i$, are the first-ones – in energy or temperature – to be rejected by quantum effects. At zero temperature (at the lowest energies) the bound to chaos is dominated by the longest length scales. Quantum effects smooth out the fine details that correspond to the short length scale contributions to chaos and progressively select the smallest Lyapunov exponents corresponding to the longest length scales, as shown pictorially in Fig.6b.

The quantum model (at this stage without the perturbation $\gamma V$) can be solved exactly and one can explicitly write the eigenstates of the Laplace-Beltrami as combinations of Legendre functions. We refer to App.E for all the details. The associated quantum spectrum is characterized by the gaps in Eq.(63). In Fig.7 we show as illustrative example the case with $n = 2$ different curvatures fixing $R_2 = 2R_1 = 2$. We plot three different wave-functions obtained decreasing the dimensionless energies: one above the two gaps $h_n^{(0)} \gg \delta_1, \delta_2$ [panel (a)], one for an intermediate energy $\delta_1 > h_n^{(0)} > \delta_2$ [panel (b)] and the last-one for $h_n^{(0)} \ll \delta_2$ [panel (c)].

# 7 Towards macroscopic systems

As mentioned in the introduction, our future goal is to study macroscopic systems, as depicted in Fig.8. In phase space, a system of $N$ hard spheres is as a point moving freely inside a

(a) Macroscopic billiard          (b) Macroscopic manifold

Figure 8: Examples of macroscopic systems that can be represented as free propagation on configuration space. (a) *Hard spheres as a macroscopic billiard.* Its structure corresponds to a polytope, with a continuous distribution of characteristic lengths: going from very short (of the order of the distance between particles) to very long (corresponding to collective rearrangements, comparable to the size of the system). (b) *Spin-liquids as a macroscopic manifold.* Example of a spin system of fixed norm $S$ with a Heisenberg interaction on the kagome lattice with fully connected couplings connecting all hexagons (shown only once) [15]. Classical ground states satisfy the constraint of vanishing total spin on each hexagon.

region bounded by the no-overlap condition for the spheres: it is a $3N$-dimensional billiard, whose structure is well-understood [14]. On the other hand, spin models with strong antiferromagnetic interaction on frustrated lattices can be characterized by highly degenerate classical ground states satisfying the constraint of zero total spin on each plaquette. This, together with the condition of the fixed norm of the individual spins, defines a curved manifold.

The quantum mechanisms illustrated in Section 5 can be easily generalized to $N$ dimensional manifolds, as done in App.F. In what follows, we describe in some more detail the two possible connections of the above arguments to hard-sphere or spin-liquids.

**Hard spheres at amorphous jamming point**    As mentioned above, a set of $N$ hard spheres in $d$ dimensions may be seen as a $dN$-dimensional billiard. Its shape has been studied by Brito and Wyart ([14], see also [31]). It may be estimated by the eigenvalues $(I_\alpha)$ of the correlations $\langle x_i x_j \rangle$ of the spheres evolving during a long-time interval. The $\sqrt{I_\alpha}$ will give us an estimate of the characteristic phase-space lengths in each direction. Near jamming – the highest density reached after a fast compression – these eigenvalues are distributed according to a function $D(I)$ that goes all the way up to infinity, meaning that the phase-space billiard has sides of lengths that are widely distributed, and, in the thermodynamic limit, without upper bound. One may thus expect that making the de Broglie length longer, all directions that are $I_\alpha < (\ell_{dB})^2$ are excluded by quantum effects, while those that are $I_\alpha \gg (\ell_{dB})^2$ are essentially classical. The Lyapunov exponent should be dominated at each temperature by a length $I \sim (\ell_{dB}(T))^2$, and one may expect it to be of order $\sqrt{\frac{T}{\ell_{dB}(T)}}$. This situation shall correspond to the model with many curvatures (see Section 6), each playing the dominant role as a source of chaos at a given temperature.

**Spin liquids**    Consider a set of spherical spins $\vec{S}_i$ with $d$ components and $|\vec{S}_i| = S$ at each vertex of a Kagome lattice, see Fig.8b. The spins live on the manifold described by the spherical constraints plus $\sum_{i \in plaquette} \vec{S}_i = 0$. A classical Hamiltonian may be written simply as the free motion on the manifold determined by these constraints. The case $d = 2$ was stud-

ied classically by Bilitewski, Bhattacharjee and Moessner in Refs. [15,32] with one (probably unimportant) difference: the surface of the sphere is itself a phase space, rather than configuration space: this is because the dynamics are precessional. In this context, one may ask whether quantum effects promote avoidance of the curved regions also in phase-space. These regions shall correspond to defects in configuration space, as opposed to the spin waves.

# 8 Outlook and conclusions

In this work, we have studied the quantum bounds at low temperature in the context of a free motion on curved manifolds. We have shown that the quantum viscosity and Lyapunov exponent are a universal function of the ratio of the smallest characteristic length of the problem and the thermal de Broglie length, the measure of the extent of the quantum fluctuations. In a classical approximation, the bounds are violated exactly when these two lengths are of the same order. We have focused on the quantum Lyapunov exponent and identified at least three mechanisms (size, effective potentials and wave-packet spreading) enforcing the bound. It is interesting to see that there are different ways quantum mechanics acts at low temperature. Finally, we have studied a model characterized by a hierarchy of divergent length scales and discussed how quantum effects progressively smooth out the smallest scales that contribute the most to chaos. Accordingly, the bounds arise in the limit of zero temperature as a collective (long-wave-length) effect.

Our findings prepare the ground for a series of extensions and challenges that would be worth exploring.

A first and useful extension concerns the bounds in phase-space, where the ground-state energy is a manifold in phase – rather than configuration – space. This setting naturally describes spin Hamiltonian dynamics at low energies (for instance the low energy limit of spin liquids) where the scaling of the Lyapunov exponent $\lambda \sim \sqrt{T}$ has already been found numerically [15].

The real challenge is to extend this approach to fermionic systems (SYK like) [7,33], where one may be able to understand – within this simple picture – the physical peculiarities of the models saturating the bounds.

Finally, let us mention that in the setting of Riemannian geometry, the problem of bounds on transport (diffusion around a handle), and chaos, is amenable to the study by mathematicians.

# Acknowledgements

We wish to thank C. Murthy, X. Turkeshi and A. Nahum for useful discussions.

**Funding information** SP and JK are supported by the Simons Foundation Grant No. 454943.

# A  Details on the classical dynamics on the pseudosphere with the cylinder

The classical dynamics is described by the Hamilton-Jacobi equations given by the Hamiltonian (43). They read

$$
\begin{cases}
\dot{\phi} = \frac{1}{\mu R^2} \frac{I^2}{\sinh^2 \tau^2} \\
\dot{I} = \sum_p \gamma_p \, p \sin(p\phi) \left[\tanh(\tau/2)/x\right]^p & \text{for} \quad \tau < \tau_x \\
\dot{\tau} = \frac{1}{\mu R^2} p_\tau \\
\dot{p}_\tau = \frac{1}{\mu R^2} \frac{I^2 \cosh \tau}{\sinh \tau^3} - \frac{1}{2} \sum_p \gamma_p \, \cos(p\phi) \, p \, \frac{[\tanh(\tau/2)]^{p-1}}{x^p \cosh^2(\tau/2)}
\end{cases}
\tag{A.1a}
$$

$$
\begin{cases}
\dot{\phi} = \frac{1}{\mu R^2} \frac{I^2}{\sinh^2 \tau_x^2} \\
\dot{I} = \sum_p \gamma_p \, p \sin(p\phi) & \text{for} \quad \tau_x < \tau < \tau_x + \tau_L \, . \\
\dot{\tau} = \frac{1}{\mu R^2} p_\tau \\
\dot{p}_\tau = 0
\end{cases}
\tag{A.1b}
$$

To solve them, we fix an initial energy $E$ and sample the initial conditions according to the equilibrium distribution of the free Hamiltonian. For the coordinates, one has

$$
P(\phi) = \frac{1}{2\pi} \, ,
\tag{A.2a}
$$

$$
P(\tau) = \frac{1}{\cosh \tau_x - 1 + \tau_L \sinh \tau_x}
\begin{cases}
\sinh \tau & \text{for} \quad \tau < \tau_x \\
\sinh \tau_x & \text{for} \quad \tau_x < \tau < \tau_x + \tau_L
\end{cases} \, .
\tag{A.2b}
$$

The momenta are extracted uniformly both in the cylinder and in the pseudosphere. In the latter, we transform back and forward to the Poincaré disk [with coordinates $(y_1, y_2)$ such that $\sqrt{y_1^2 + y_2^2} = \tanh \tau/2$] where the Hamiltonian reads $H_0 = \frac{(1 - y_1^2 - y_2^2)^2}{4}(p_1^2 + p_2^2)$. Here, we extract uniformly $p_1$ ad $p_2$ rescaling them, such that the total energy is $H_0 = E$. Then, the equations of motion (A.1) are integrated numerically with a fourth-order Runge-Kutta algorithm, fixing the relative and absolute error to $10^{-12}$.

The integrability breaking term $\gamma V(\tau, \phi)$ is chosen such that the relative error on the energy $E$ is below 2%, i.e. $|\gamma V(\tau, \phi)|/E < 0.02$. We verify that the choice of $V(\tau, \phi)$ does not affect our results. In particular, we check that the long-time distribution of the observables, e.g. the radial coordinate $\tau$, corresponds to the equilibrium distribution (A.2). This is shown for a typical perturbation in Fig.9, where we compare the long-time average (up to a time $T_f$) with the predicted equilibrium one, finding excellent agreement.

The classical square commutator (45)

$$
c_{cl}(t) = \langle\!\langle \{\tau(t), \tau(0)\}^2 \rangle\!\rangle_E = \left\langle\!\!\left\langle \left(\frac{d\tau(t)}{dp_\tau(0)}\right)^2 \right\rangle\!\!\right\rangle_E
\tag{A.3}
$$

is computed in the following way: For each initial condition with momentum $p_\tau(0)$, we consider a different trajectory changing only the momentum by $p_\tau'(0) = p_\tau(0) + \epsilon$ (with $\epsilon = 10^{-6}$). We then compute the difference $(\tau(t) - \tau'(t))^2/\epsilon^2$. The extent the infinitesimal displacement $\epsilon$ fixes the saturation value of the classical square-commutator, that saturates at large times to $\sim \epsilon^{-2}$, as shown also in Fig.3. We then average over different initial conditions at energy $E$, typically $\sim 100$.

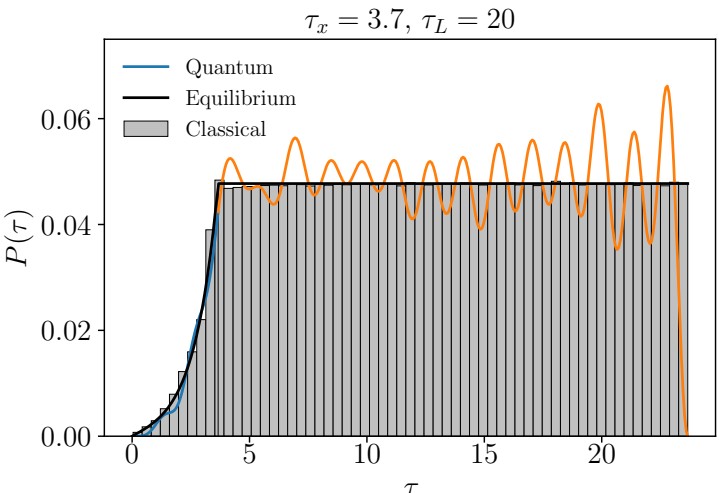

Figure 9: Comparison between the classical (A.2) and the quantum distribution of the radial coordinate $P(\tau)$ at high energies. The boxed grey area represents a histogram of the classical trajectory averaged up to time $T_f = 1000$ and over 50 initial conditions sampled according to the equilibrium distribution. The blue and the orange lines correspond to the modulus square of the quantum wave functions centred around $n_0 = 500$ ($h_{n_0} = 23$) of 30 states. The classical energy is $E = 2.2$. The parameters of the perturbations are the same as in Fig.3 and Fig.10.

# B  Properties of Legendre functions

In this appendix we summarize the properties – relevant for our discussions – of the associated Legendre functions, following Refs. [34, 35]. The general Legendre equation reads

$$(1-z^2)\,y'' - 2z\,y' + \left[\nu(\nu+1) - \frac{\mu^2}{1-z^2}\right]y = 0\,, \tag{B.1}$$

where $\mu$, $\nu$ and $z$ can be complex variables. We are interested in solutions of the equation that correspond to real values of the variable $|z| > 1$. The linearly independent functions

$$P_\mu^\nu(z) = \left(\frac{z+1}{z-1}\right)^{\mu/2} \mathbf{F}\left(\nu+1, -\nu; 1-\mu; \tfrac{1}{2} - \tfrac{1}{2}z\right), \tag{B.2}$$

$$Q_\mu^\nu(z) = e^{\mu\pi i}\,\frac{\pi^{1/2}\Gamma(\nu+\mu+1)\left(z^2-1\right)^{\mu/2}}{2^{\nu+1}z^{\nu+\mu+1}}\,\mathbf{F}\left(\tfrac{1}{2}\nu + \tfrac{1}{2}\mu + 1, \tfrac{1}{2}\nu + \tfrac{1}{2}\mu + \tfrac{1}{2}; \nu + \tfrac{3}{2}; \frac{1}{z^2}\right), \tag{B.3}$$

with $\mathbf{F}(a, b, c, d)$ the Olver's hypergeometric function, are solutions of the differential equation (B.1) and are called *associated Legendre functions* (or spherical functions) of the *first* and *second kind* respectively of *degree $\nu$* and *order $\mu$*.

If $\nu \pm \mu$ is not an integer, Eq.(B.1) admits as solutions $P_\nu^{\pm\mu}(z)$, $Q_\nu^{\pm\mu}(z)$, $P_{-\nu-1}^{\pm\mu}(z)$ and $Q_{-\nu-1}^{\pm\mu}(z)$. Nonetheless, two linearly independent solutions can always be found. In particular, the Legendre functions are related by several relations, like the Whipple formula[8] or, for

---

[8]See Eqs.(13-14) of Chapter 3.3.1 in Ref. [35]

instance [9],

$$P_\mu^{-\nu-1}(x) = P_\mu^\nu(x) \,, \tag{B.4a}$$

$$e^{i\pi\mu}\Gamma(\nu+\mu+1)Q_{-\mu}^\nu(z) = e^{-i\pi\mu}\Gamma(\nu-\mu+1)Q_\mu^\nu(z) \,, \tag{B.4b}$$

$$Q_\mu^\nu(z) = \frac{e^{i\pi\mu}}{2\pi\sin(\mu\pi)}\left(P_\mu^\nu(z) - \frac{\Gamma(\nu+\mu+1)}{\Gamma(\nu-\mu+1)}P_{-\mu}^\nu(z)\right) \,, \tag{B.4c}$$

$$P_\mu^\nu(z) = \frac{e^{-i\mu\pi}}{\pi\cos(\pi\mu)}\left(Q_\mu^\nu(z)\sin[\pi(\nu+\mu)] - Q_\mu^{-\nu-1}(z)\sin[\pi(\nu-\mu)]\right) \,. \tag{B.4d}$$

The Legendre functions also obey recursion relations. In particular, the derivatives can be written recursively as [36]

$$(z^2-1)\frac{d}{dz}P_\mu^\nu(z) = \sqrt{z^2-1}P_{\mu+1}^\nu(z) + \mu z P_\mu^\nu(z) \,, \tag{B.5a}$$

$$(z^2-1)\frac{d}{dz}Q_\mu^\nu(z) = \sqrt{z^2-1}Q_{\mu+1}^\nu(z) + \mu z Q_\mu^\nu(z) \,. \tag{B.5b}$$

For $z \to 1^+$, the Legendre functions can be expanded as [10]

$$P_\mu^\nu(z) \sim \frac{1}{\Gamma(1-\mu)}\left(\frac{2}{z-1}\right)^{\mu/2} \,, \quad \mu \neq 1,2,3,\dots \,, \tag{B.6a}$$

$$Q_\mu^\nu(z) \sim \frac{e^{i\pi\mu}}{2}\Gamma(\mu)\left(\frac{2}{z-1}\right)^{\mu/2} \,, \quad \text{Re}(\mu) > 0, \nu+\mu \neq -1,-2,\dots \,, \tag{B.6b}$$

$$Q_{\mu=0}^\nu(z) = -\frac{1}{2}\ln\left(\frac{z}{2}-1/2\right) + -\gamma - \psi(\nu+1) + O(z-1) \,, \quad \nu \neq -1,-2,-3,\dots \,, \tag{B.6c}$$

while for $z \to \infty$ they can be expanded as [11]

$$P_\mu^\nu(z) \sim \frac{\Gamma\left(\nu+\frac{1}{2}\right)}{\pi^{1/2}\Gamma(\nu-\mu+1)}(2z)^\nu \,, \quad \text{Re}(\nu) > -\frac{1}{2}, \mu-\nu \neq 1,2,3,\dots \tag{B.7a}$$

$$P_\mu^\nu(z) \sim \frac{\Gamma\left(-\nu-\frac{1}{2}\right)}{\pi^{1/2}\Gamma(-\nu-\mu)}(2z)^{-\nu-1} \,, \quad \text{Re}(\nu) < -\frac{1}{2}, \mu-\nu \neq 1,2,3,\dots \tag{B.7b}$$

$$Q_\mu^\nu(z) \sim \pi^{1/2}\frac{e^{i\mu\pi}\Gamma(\nu+\mu+1)}{\Gamma\left(\nu+\frac{3}{2}\right)}\frac{1}{(2x)^{\nu+1}} \,, \quad \nu \neq -\frac{3}{2},-\frac{5}{2},-\frac{7}{2},\dots \,. \tag{B.7c}$$

The integral representation reads

$$P_m^\nu(z) = \frac{2^\mu(z^2-1)^{-\mu/2}}{\sqrt{\pi}\,\Gamma(1/2-\mu)}\int_0^\pi\left[z+\sqrt{z^2-1}\cos\phi\right]^{\nu+\mu}(\sin\phi)^{-2\mu}d\phi \,, \quad \text{Re}(\mu) < 1/2 \tag{B.8a}$$

$$Q_\mu^\nu(z) = e^{\mu\pi i}\frac{(z^2-1)^{-\mu/2}\Gamma(\nu+\mu+1)}{2^{\nu+1}\Gamma(\nu+1)}\int_0^\pi[z+\cos\phi]^{\mu-\nu-1}(\sin\phi)^{2\nu+1}d\phi \,, \tag{B.8b}$$

$$\text{Re}(\nu) > -1 \,, \quad \text{Re}(\nu+\mu+1) > 0 \,. \tag{B.8c}$$

---

[9]See Eqs.(1-10) of Chapter 3.3.1 in Ref. [35]
[10]See Eqs.(14.8.7),(14.8.9) and (14.8.11) in Ref. [34], where they are expressed in terms of $\boldsymbol{Q}_\nu^\mu(z) \equiv e^{-i\pi\mu}Q_\nu^\mu(z)/\Gamma(\nu+\mu+1)$, such that $\boldsymbol{Q}_\nu^\mu(z) = Q_\nu^{-\mu}(z)$.
[11]See Eqs.(14.8.2) and (14.8.15) in Ref. [34]

For $\mu = m \in \mathbb{Z}$ and $z = \cosh\tau$, the integral representations (B.8) become

$$P_m^{\nu}(\cosh\tau) = \frac{\Gamma(\nu + m + 1)}{2\pi\Gamma(\nu + 1)} \int_0^{2\pi} (\cosh\tau + \sinh\tau\,\cos\phi)^{\nu}\, e^{im\phi}\, d\phi \qquad (B.9a)$$

$$Q_{m=0}^{\nu}(\cosh\tau) = 2^{-1/2} \int_{\tau}^{\infty} d\tau' \frac{e^{-(\nu+1/2)\tau'}}{\sqrt{\cosh\tau' - \cosh\tau}}\ . \qquad (B.9b)$$

The semi-classical WKB formulae for $\nu \gg 1$ of the Legendre functions are obtained by applying the saddle-point method to integral representations. By defining $\nu = -\frac{1}{2} + i\rho$, one has for $\rho \to \infty$ with $\rho \gg m$ [12]

$$P_m^{\nu=-\frac{1}{2}+i\rho}(\cosh\tau) \sim i^m \frac{\cos(\rho^{-}\tau\frac{\pi}{4} - m\frac{\pi}{2})}{\sqrt{2\pi\rho^{-}\sinh\tau}}\ , \qquad (B.10)$$

$$Q_m^{\nu=-\frac{1}{2}+i\rho}(\cosh\tau) \sim \sqrt{\frac{\pi}{2\rho\sinh\tau}}\, e^{-i(\rho\tau + \pi/4)}\ . \qquad (B.11)$$

# C  Details on the quantum dynamics on the pseudosphere with the cylinder

## C.1  Eigenfunctions and eigenvalues

We start by solving the standard Schrödinger equation

$$\hat{H}_0\,\psi(\tau, \psi) = E\,\psi(\tau, \psi)\ , \qquad (C.1)$$

with the unperturbed Hamiltonian defined in Eqs.(46),(48). We first use the separation of variables and write the wave-function as

$$\psi(\tau, \phi) = \frac{e^{im\phi}}{\sqrt{2\pi}}\, F(\tau)\ , \qquad m = 0, \pm 1, \pm 2, \dots\ , \qquad (C.2)$$

where the discreteness of $m$ comes from requiring the periodicity in $\phi$ and the function $F(\tau)$ satisfies

$$\left[ \frac{1}{\sinh\tau} \frac{\partial}{\partial\tau} \left( \sinh\tau \frac{\partial}{\partial\tau} \right) - \frac{m^2}{\sinh\tau^2} + h \right] F_A(\tau) = 0 \quad \text{for} \quad \tau < \tau_x\ ,$$

$$\left[ \frac{\partial^2}{\partial\tau^2} - \frac{m^2}{\sinh\tau_x^2} + h \right] F_B(\tau) = 0 \quad \text{for} \quad \tau_x < \tau < \tau_x + \tau_L\ , \qquad (C.3a)$$

where we have defined the adimensional energies

$$h = \frac{2\mu R^2}{\hbar^2} E\ . \qquad (C.4)$$

For $\tau < \tau_x$, the equation corresponds to the *Legendre equation* in Eq.(B.1) with $z = \cosh\tau$ of order $m$ and degree $\ell$ upon identifying $\ell(\ell + 1) = -h$. This equation admits as solution the Legendre functions of first and second kind, i.e.

$$P_m^{\ell}(z)\ , \quad Q_m^{\ell}(z)\ , \qquad \text{for} \quad |z| > 1\ . \qquad (C.5a)$$

---

[12]See Eqs.(G17),(G18) in Ref. [11].

The Legendre functions of second kind $Q_m^\ell(z)$ is divergent for $|z| \to 1$, hence the solution for $0 \le \tau < \tau_x$ is given by

$$F_A(\tau) = P_m^\ell(\cosh \tau) \,, \tag{C.6}$$

where

$$h = -\ell(\ell+1)\,, \quad \ell = -\frac{1}{2} \pm i\rho \,, \quad \rho = \sqrt{h - \frac{1}{4}}\,. \tag{C.7}$$

For $\tau_x < \tau < \tau_x + \tau_L$, we identify $\kappa^2 = h - \frac{m^2}{\sinh^2 \tau_x}$. For $\kappa^2 > 0$ the solution reads

$$F_B(\tau) = C \cos[\kappa(\tau - \tau_x)] + D \sin[\kappa(\tau - \tau_x)]\,, \tag{C.8}$$

where the constants $C$ and $D$ are determined by imposing the continuity of the wave-function and of its derivative at $\tau = \tau_x$. The condition is set on the normalized wave-function (47):

$$\Phi_A(\tau_x) = \Phi_B(\tau_x) \quad \longrightarrow \quad C = P_m^\ell(\cosh \tau_x) \tag{C.9a}$$

$$\frac{d\Phi_A}{d\tau}\Big|_{\tau=\tau_x} = \frac{d\Phi_B}{d\tau}\Big|_{\tau=\tau_x} \quad \longrightarrow \quad \kappa D \sqrt{\sinh \tau_x} = \frac{1}{2}\frac{\cosh \tau}{\sqrt{\sinh \tau}} P_m^\ell + \sqrt{\sinh \tau}\frac{dP_m^\ell}{dz}\frac{dz}{d\tau}\Big|_{\tau=\tau_x}\,, \tag{C.9b}$$

with $z = \cosh \tau$ the argument of the Legendre function. The derivatives of the Legendre functions can be written in a recursive way as in Eq.(B.5a) [36]. From these, the continuity of the first derivative (C.9b) for $z = \cosh \tau_x$ implies

$$D = \frac{1}{\kappa}\left[ P_{m+1}^\ell(\cosh \tau_x) + \left(m + \frac{1}{2}\right)\coth \tau_x\, P_m^\ell(\cosh \tau_x)\right]\,.$$

We now find the quantization condition by imposing that the solution vanishes at the boundaries, i.e.

$$F(\tau_x + \tau_L) = F_B(\tau_x + \tau_L) = 0\,, \tag{C.10}$$

leading to

$$C \cos[\kappa \tau_L] + D \sin[\kappa \tau_L] = 0\,.$$

Notice that the discretization condition does not admit solution for $\kappa^2 < 0$, where one should substitute "cos, sin, tan" in the previous equations (C.8) with "cosh, sinh, tanh".

Summarizing, the normalized *eigenstates* $\Phi_n = g^{1/4}\Psi_n$ read

$$\Phi_n(\tau,\phi) = \frac{e^{im_n \phi}}{\mathcal{N}_n} \tag{C.11}$$

$$\times \begin{cases} \sqrt{\sinh \tau}\, P_{m_n}^{\ell_n}(\cosh \tau) & \text{for} \quad \tau < \tau_x\,, \\ \sqrt{\sinh \tau_x}\,(C_n \cos[\kappa_n(\tau - \tau_x)] + D_n \sin[\kappa_n(\tau - \tau_x)]) & \text{for} \quad \tau_x < \tau < \tau_x + \tau_L\,, \end{cases}$$

where the coefficients read, $\forall \kappa_n \ne 0$,

$$C_n = P_m^{\ell_n}(\cosh \tau_x)\,, \tag{C.12}$$

$$D_n = \frac{1}{\kappa_n}\left[ P_{m+1}^{\ell_n}(\cosh \tau_x) + \left(m + \frac{1}{2}\right)\coth \tau_x\, P_m^{\ell_n}(\cosh \tau_x)\right]\,.$$

The quantization condition is found by solving the following condition

$$\tan(\kappa_n \tau_L) = -\frac{C_n}{D_n}\,, \tag{C.13}$$

while the adimensional *eigenvalues* $h^{(0)}$ are related to the degree $\ell_n$ and the momentum $\kappa_n$ in (C.11) by

$$\ell_n = -\frac{1}{2} + \sqrt{\frac{1}{4} - h_n^{(0)}}\,, \quad h_n^{(0)} = \kappa_n^2 + \frac{m_n^2}{\sinh^2 \tau_x}\,. \tag{C.14}$$

The classical limit $\hbar \to 0$ is retrieved for $n \to \infty$, with $1/\sqrt{n}$ as a semi-classical parameter. To be in this regime, it is enough to choose $\tau_x \gg 1$, $\tau_L \gg 1$ leading to a denser spectrum via the quantization condition. This is equivalent to avoiding the trivial effect due to the small system size [see the discussion on the mechanism (1) at the beginning of Section 5]. See the plot of the wave-functions (C.11) in Fig.4 and the relative discussion.

## C.2 Quantum dynamics

We now address the quantum dynamics. We proceed as in the classical case: we first add the perturbation, we check how this affects the equilibration of the system and then study the effects on the chaotic dynamics. We break the conservation of the angular momentum $\hat{I}$ considering the following Hamiltonian

$$\hat{H} = \hat{H}_0 + \sum_p \gamma_p V_p(\hat{\tau}, \hat{\phi})\,, \tag{C.15}$$

where $V_p(\hat{\tau}, \hat{\phi})$ is the same as in Eq.(44) where now the classical variables $\tau$ and $\phi$ are substituted by the operators. The perturbed Hamiltonian is written as a matrix in the unperturbed basis (C.11) as $H_{\alpha\beta} = E_\alpha^{(0)} \delta_{\alpha\beta} + V_{\alpha\beta}$, where the matrix elements of the perturbation $\hat{V}$ are

$$V_{\alpha\beta} = \sum_p \gamma_p \int d\tau d\phi\, \Phi_\alpha^*(\tau, \phi) V_p(\tau, \phi) \Phi_\beta(\tau, \phi)\,. \tag{C.16}$$

Notice that for each $p$ the perturbation connects only eigenstates whose angular momentum differs by $p$, i.e. $|m_\alpha - m_\beta| = p$. This is the reason why several terms in $p$ are needed to ensure that the perturbation is not block diagonal. In our numerical simulations, we find that it is enough to consider few values of $p$. The umperturbed basis is fixed by solving numerically the condition (C.13) for $N_{cut} \gg 1$ number of states, that we increase up to the convergence of the results. Then, we compute numerically the matrix elements in Eq.(C.16), leading to a $N_{cut} \times N_{cut}$ matrix $\hat{H}$. We diagonalize it and find the corresponding spectrum $E_n = \frac{\hbar^2}{2\mu R^2} h_n$ and eigenstates. The comparison between the unperturbed and perturbed solutions is shown in Fig.10, where the different spectra $h_n^{(0)}$ and $h_n$ are displayed in panels (a,d), while in panels (b,e) we illustrate the expectation value of the angular momentum $\langle n|\hat{I}|n\rangle$ of each eigenstate. This plot is equivalent to a Peres lattice [37], a standard tool for visualizing regularity or chaoticity of the spectrum [13]. Notice that the perturbation is chosen to be small, such that, in the energy region of interest, it does not change the spectrum, yet it breaks the conservation of the angular momentum, as shown in panel (b). The Peres lattice becomes particularly irregular for small values of $\langle n|\hat{I}|n\rangle$, signalling that chaos is induced by the geometry of the pseudosphere.

To inspect the quality of our integrability breaking, we study the distribution of the angular momenta $\langle n|\hat{I}|n\rangle$ as a function radial quantum numbers $\kappa_n = \sqrt{h_n - \langle n|\hat{I}|n\rangle^2/\sinh^2 \tau_x}$ [cf. Eq.(C.14)] in Fig.10c. Since the energy is a sum of these two contributions, the two are

---

[13]Whenever a system is integrable, observables in the energy eigenbasis form a regular pattern, the so-called Peres lattice. Conversely, for chaotic spectra, observables generally lie in an "irregular" fashion.

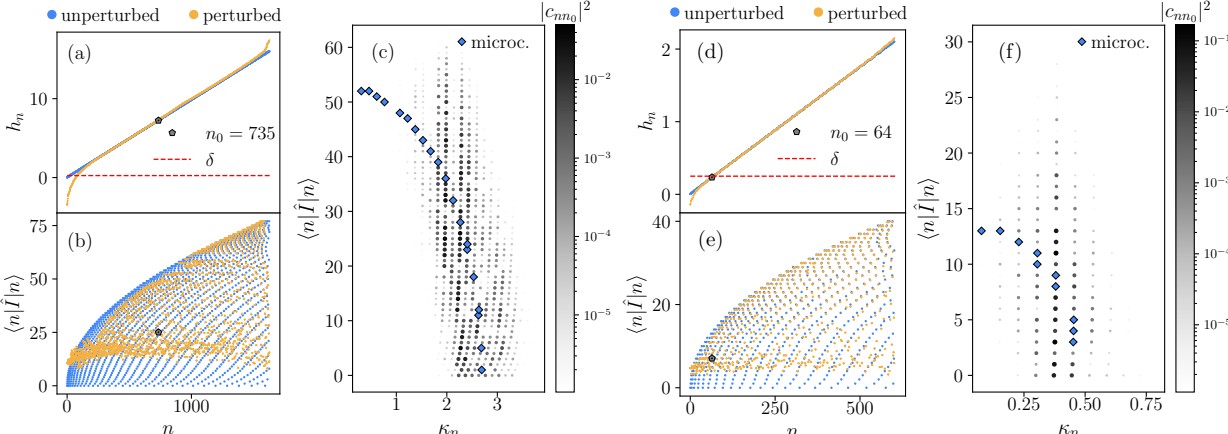

Figure 10: Chaotic spectrum for high energy states. (a,b,d,e) Comparison between the unperturbed dimensionless spectrum (blue) and the perturbed one (orange) of the Hamiltonian (C.15). (a,d) Dimensionless spectrum. (b,e) Expectation value of the angular momentum $\langle n|\hat{I}|n\rangle$. (c,f) Equilibrium distribution of angular momenta vs radial quantum numbers $\kappa_n$. (Blue diamonds) microcanonical expectation value of the unperturbed eigenstates (C.11), (grey dots) distribution of the overlaps for the eigenstate $|n_0\rangle$. Parameters: (a-c) High energy states: $n_0 = 735$, window obtained with 20 eigenstates with an energy difference of $\Delta h^{(0)} = 0.04$. Geometry: $\tau_L = 20$, $\tau_x = 3.7$. Perturbation with $p = (10, 7, 5, 2)$ with $\gamma_p = (-0.8, 1.6, 2, -3)$. (d-f) States close to the gap: $n_0 = 64$, window obtained with 10 eigenstates with an energy difference of $\Delta h^{(0)} = 0.04$. Geometry: $\tau_L = 40$, $\tau_x = 4$. Perturbation with $p = (10, 7, 5, 2)$ with $\gamma_p = (-0.5, 1, 1.5, -2.5) \times 10^{-1}$.

expected to be distributed in a smooth curve at equilibrium. We compare the result of the "microcanonical" distribution of $\hat{H}_0$ (a small energy window of width $\Delta h^{(0)}$) with the distribution of a single eigenstate $|n_0\rangle$ of $\hat{H}$. The latter is obtained by looking at the distribution of the overlaps between $|n_0\rangle$ and the unperturbed basis $|n^{(0)}\rangle$, i.e. $|c_{nn_0}|^2 = |\langle n^{(0)}|n_0\rangle|^2$. We find a good agreement between the two predictions. This is a nice illustration of the eigenstate thermalization hypothesis (ETH) [38], that states that a single chaotic eigenstate $|n_0\rangle$ "contains" all the equilibrium distributions.

We are now in position to study the dynamics and the quantum Lyapunov exponent. We consider the *microcanonical* version of the square-commutator in Eq.(25), see, e.g., Ref. [28]. We focus on the evolution of

$$c(t) = -\langle n_0 | [\hat{\tau}(t), \hat{\tau}(0)]^2 | n_0 \rangle \,, \tag{C.17}$$

where the system is initialized in an eigenstate $|n_0\rangle$ of the perturbed Hamiltonian $\hat{H}$ (C.15). In all the examples below, $c(t)$ is evaluated numerically by writing explicitly the matrix elements of $\tau_{\alpha\beta} = \delta_{m_\alpha m_\beta} \int d\tau \, \Phi_\alpha^*(\tau) \tau \Phi_\beta(\tau, \phi)$ in the umperturbed basis (C.11) and by computing exactly Eq.(C.17) with vector-matrix multiplications. First of all, we study the classical limit $n_0 \gg 1$ at high energy $h_{n_0} \gg 1$. In Fig. 5, we show the evolution of the square commutator with different initial conditions increasing in $n_0$. The curves show that $c(t)$ grows exponentially with twice the classical total Lyapunov exponent [cf. Eq.(42)] at the corresponding energy. Due to quantum interference, this growth holds only in a time window – the Lyapunov regime – that ends at the Ehrenfest time $T_{Ehr} \propto \log \hbar_{\text{eff}}^{-1} \propto \log n_0$. Indeed, the Lyapunov regime is longer for higher energies, as shown in Fig.5.

We now contrast this behaviour with semiclassical states $n_0 \gg 1$ that, instead, possess energies right below the gap $h_{n_0} \lesssim \delta$. First of all, we verify the impact of the integrability breaking at low energies in Fig.10. We check that the perturbation $\hat{V}$ does not affect too much the spectrum [panel (d)], yet it mixes sufficiently enough the angular momentum [panel (e)]. In Fig.10f, we find agreement between the microcanonical (unperturbed) angular momentum distribution with the one of a single eigenstate at energies below the gap. Despite all these properties, the microcanonical square commutator (C.17) initialized with $h_{n_0} \lesssim \delta$ does not display an exponential regime, in agreement with our picture for the bound to chaos. See Fig.5b as an illustrative example, wherein the inset we show that $c(t)$ grows only polynomially fast in time.

# D  Dynamics on the pseudosphere with a boundary

In this Appendix, we study the classical and quantum dynamics of a free particle on a finite portion of the pseudosphere, described by the metric

$$ds^2 = R^2(d\tau^2 + \sinh^2\tau\, d\phi^2)\,. \tag{D.1}$$

The surface is closed by adding a boundary in the form of an infinite wall at $\tau = \tau_x$. Firstly, we look at the solution of the classical model and see how exponential divergence of trajectories arise. Secondly, we solve the quantum problem for $\tau_x \gg 1$ and we find that the system behaves like a free particle in a large box with the length depending on the angular momentum. Understanding this situation is relevant for understanding the bound to chaos. In fact, the quantum gap $\Delta$ [cf. Eq.(51)] – responsible for the slowing down of the particle of mechanism (2) – disappears in the absence of the cylinder or with Dirichlet boundary conditions. The detailed study of this problem leads to the mechanism (3), explained in Section 5.6.

## D.1  Minimal classical model for the Lyapunov exponent

We study the simplest solution of the classical chaotic Hamiltonian dynamics, described by the metric in Eq.(D.1) and the corresponding Hamiltonian

$$E = \frac{1}{2\mu R^2}\left[p_\tau^2 + \frac{I_\phi^2}{\sinh^2\tau}\right]\,.$$

The particle is initialized in some $\tau_0$ and we fix $\tau_x \gg 1$ to study only the initial free dynamics. The Hamilton-Jacobi equation of motion are

$$\begin{cases} \dot{\phi} = \frac{I_\phi^2}{\sinh^2\tau^2} \\ \dot{I}_\phi = 0 \\ \dot{\tau} = p_\tau \\ \dot{p}_\tau = \frac{I_\phi^2\cosh\tau}{\sinh\tau^3} \end{cases} \times \frac{1}{\mu R^2}\,. \tag{D.2}$$

Let us first determine the solution for the radial coordinate $\tau(t)$. From the Hamiltonian, we re-write $p_\tau = \pm\sqrt{2E\mu R^2 - \frac{I_\phi^2}{\sinh^2\tau}}$ and, using the third equation in (D.2), we need to solve

$$\frac{d\tau}{\sqrt{2E\mu R^2 - \frac{I_\phi^2}{\sinh^2\tau}}} = \mu R^2\, dt\,. \tag{D.3}$$

We consider the solution propagating outward and, integrating both sides from $t_0$ to $t$ and $\tau_0$ to $\tau(t)$, we obtain

$$\log \frac{\cosh \tau(t) + \sqrt{\sinh^2 \tau(t) - I_\phi^2/2E\mu R^2}}{\cosh \tau_0 + \sqrt{\sinh^2 \tau_0 - I_\phi^2/2E\mu R^2}} = \sqrt{\frac{2E}{\mu R^2}}(t - t_0) \,, \tag{D.4}$$

where we notice that $\sqrt{\frac{2E}{\mu R^2}} = \lambda_c$ is exactly the classical Lyapunov exponent [cf. Eq.(41)]. We denote $K_0$ the denominator of the logarithm in the previous equation, i.e. $K_0 = \cosh \tau_0 + \sqrt{\sinh^2 \tau_0 - I^2/2E\mu R^2}$. Eq.(D.4) admits as solution

$$\cosh \tau(t) = \frac{1}{2} K_0 e^{\lambda_c(t-t_0)} + \frac{1}{2K_0}\left(1 + \frac{I_\phi^2}{2E\mu R^2}\right) e^{-\lambda_c(t-t_0)} \,. \tag{D.5}$$

In the limit $t \gg 1$, the radial coordinate grows linearly in time

$$\tau(t) = \log K_0 + \lambda_c(t - t_0) \,. \tag{D.6}$$

Notice that in the limit $I_\phi^2 \ll 2E\mu R^2$ one has $\log K_0 \to \tau_0$. We can now study the solution for $\phi(t)$. The first line of Eq.(D.2) yields

$$d\phi = \frac{I_\phi^2}{\mu R^2}\frac{dt}{\sinh^2 \tau(t)} = \frac{I_\phi^2}{\mu R^2}\frac{dt}{\cosh^2 \tau(t) - 1} \,. \tag{D.7}$$

By substituting the solution (D.5) and integrating, one gets

$$\phi(t) = \left[\phi(0) + I_\phi\left(\arctan \phi(0) - \arctan \phi(t)\right)\right] \mod 2\pi \,, \tag{D.8}$$

$$\text{with} \quad \phi(t) = \frac{I_\phi + 2\mu E R^2/I_\phi(K_0^2 e^{2\lambda_c(t-t_0)} - 1)}{2\sqrt{2E\mu R^2}} \,.$$

In the limit $t \gg 1$ and $I_\phi^2 \ll 2E\mu R^2$, the angle is constant

$$\phi(t) = \phi(0) \,. \tag{D.9}$$

Let us now discuss how equations Eqs.(D.5),(D.8) lead to the exponential divergence of nearby trajectories. On curved surfaces, the finite Lyapunov exponent comes from geometry: the structure of the metric is crucial. The linear displacements of the polar coordinates $\delta\phi(t)$ and $\delta\tau(t)$ grow at most polynomially in time at large times. Consider two nearby trajectories with an infinitesimal angular displacement $\delta\phi(0)$: in the limit $t \gg 1$ in Eq.(D.8) one has $\delta\phi(t) \sim \delta\phi(0)$. On the other hand, if we consider two trajectories differing only by $\delta\tau_0$, from Eq.(D.6) it follows that $\delta\tau(t) \sim \delta\tau(0)\lambda_c t$.

However, distances are computed with the metric in Eq.(D.1). By choosing two trajectories with slightly different angles, all the exponential divergence in time comes from the exponential growth of $\sinh^2 \tau(t)$. More precisely, for two initial conditions with the same radial coordinate $\tau_0$ and two angles differing by $\delta\phi(0)$, the spatial separation between two geodesics is given by

$$\left(\frac{ds(t)}{ds(0)}\right)^2 = \sinh^2 \tau(t)\left(\frac{d\phi(t)}{d\phi(0)}\right)^2$$

$$= \left(\sinh \tau(t)\{\phi(t), I_\phi(0)\}\right)^2 \sim \sinh^2 \tau(t) \sim e^{2\lambda_c t} \quad \text{for} \quad t \to \infty \,,$$

where we have used the definition of the Poisson Brackets just to make the analogy with the square-commutator clear. In the third line we have used Eq.(D.6) and that angular variations remain almost constant in time, as discussed above. From this expression it is clear that all the exponential growth of the separation between two geodesics at different initial conditions comes from the way we are computing distances, hence from the metric.

## D.2 Spectrum of the pseudosphere with a boundary

We now study the Schrödinger equation

$$\hat{H}_0 \psi_n = E_n \psi_n \quad \text{or} \quad \hat{h}\psi_n = h_n \psi_n \, ,$$

where the Hamiltonian is defined in Eq.(48) with an infinite potential at $\tau = \tau_x$. The regular solution is

$$\psi_n(\tau, \phi) = \frac{e^{im\phi}}{\mathcal{N}_n} P_m^{\ell_n}(\cosh \tau) \, , \tag{D.10}$$

where $\mathcal{N}_n$ is the normalization constant and $P_m^{\ell_n}$ are the Legendre function of degree $m$ (the quantized angular momentum) and order $\ell_n$, that is related to the adimensional energy $h_n$ by

$$\ell_n = -\frac{1}{2} + i\sqrt{h_n - \frac{1}{4}} \, .$$

The quantization condition is set by the wall in $\tau_x$. The eigenenergies are determined by finding numerically, for each $m$, the zeros of the Legendre function

$$P_m^{\ell_n}(\cosh \tau_x) = 0 \, . \tag{D.11}$$

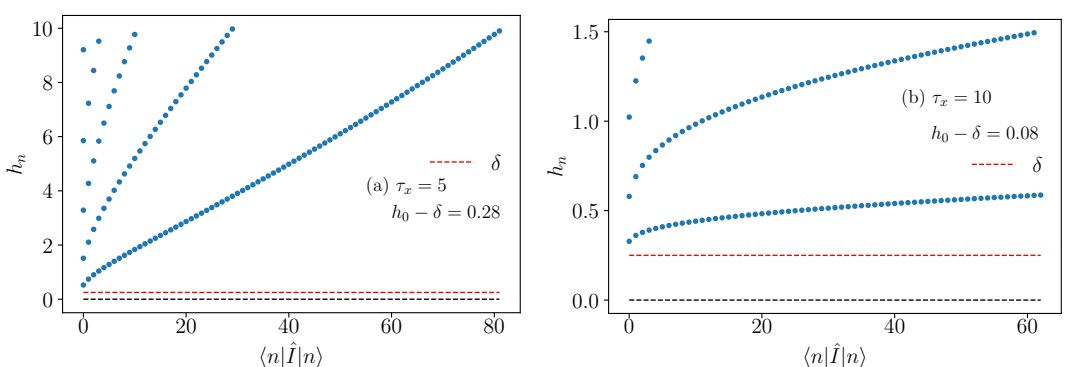

Figure 11: dimensionless spectrum obtained solving Eq.(D.11) as a function of the angular momentum of each eigenstate. We compare two different sizes (top) $\tau_x = 5$ and (bottom) $\tau_x = 10$ with approximately the same number of eigenstates. Dashed in red the value of the gap $\delta = 1/4$. By increasing $\tau_x$ the gaps between neighbouring eigenstates diminish: notice the different energy scales in the top and the bottom figures. Also, the ground state $h_0$ becomes closer to $\delta$, as emphasised in the caption of the plots.

In Fig.11, we show the adimensional energy $h_n$ as a function of the angular momentum $\hbar m_n = \langle n|\hat{I}|n\rangle$ for each eigenstate. Each branch in the plot is related to wave functions with $\nu_n = 0, 1, 2, \dots$ number of nodes (from bottom to top) for different angular momenta $m_n$. Different branches correspond to different $\nu$ and result from the quantization in the radial direction. Conversely, on each branch, we have eigenstates with different quantized angular momenta $m_n$. States with fixed $\nu$ and different $m_n$ have smaller gaps than the states with fixed $m_n$ and a different number of nodes.

To estimate the different gaps, consider the Schrödinger equation of Eqs.(49)-(50) for $\tau < \tau_x$, i.e.

$$\left[ \frac{\partial^2}{\partial^2 \tau} + \frac{1/4 - m^2}{\sinh \tau^2} \right] \Phi = \left( \frac{1}{4} - h \right) \Phi \, . \tag{D.12}$$

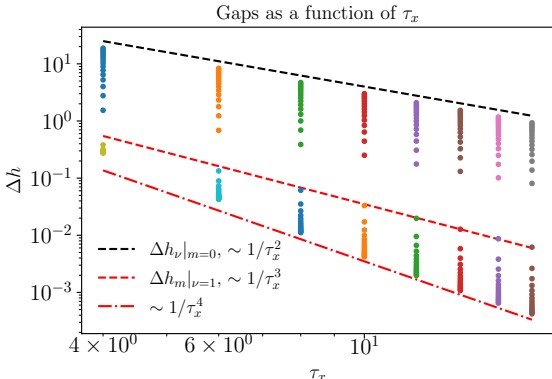

Figure 12: Exact gaps of the dimensionless spectrum obtained solving (D.11) as a function of the size of the boundary $\tau_x$. Different colours correspond to different $\tau_x$. The two different curves show the first 16 gaps obtained (top curve) for $m = 0$ varying $\nu$ (bottom curve) for $\nu = 1$ varying $m$. The dashed lines compare with the predictions of Eq.(D.15) and Eqs.(D.16)-(D.17) respectively.

The angular potential $m^2/\sinh\tau^2$ is almost zero for $m^2/\sinh^2\tau \ll 1$, i.e. $\tau \gg \operatorname{arcsinh}(m) \equiv \tau_m$. In this limit, the system can be interpreted as a free particle in a wall of length $\tau_x - \tau_m$ that admits oscillatory solutions and quantised energy eigenstates as

$$h_{\nu m} = \frac{1}{4} + \frac{\pi^2 \nu^2}{(\tau_x - \tau_m)^2} \quad \nu = 1, 2, \dots , \tag{D.13}$$

where $\nu$ is the quantum number associated to $\nu - 1$ nodes in the wave-function. In this approximation, the eigenstates can we written as

$$\Phi_{m\nu}(\tau) = \frac{e^{im\phi}}{\sqrt{\pi(\tau_x - \tau_m)}} \sin\left(\frac{\nu\pi\tau}{\tau_x - \tau_m}\right) . \tag{D.14}$$

Let us now consider states with different number of nodes and fixed $m$. Form Eq.(D.13), the gaps between consecutive $\nu$ at fixed $m$ are

$$\Delta h_\nu\big|_{m=const.} \propto \frac{2\nu + 1}{(\tau_x - \tau_m)^2} \simeq \frac{\nu}{\tau_x^2} \quad \text{for} \quad \tau_x \gg 1 . \tag{D.15}$$

This limit coincides with the semiclassical solution obtained in the limit of $h \gg 1$ and $h \gg m$, see Eq.(B.10).

On the other hand, let us fix the number of nodes $\nu$ and consider the difference in energy between two eigenstates at different $m$. From Eq.(D.13) we have

$$\begin{aligned}
\Delta h_m\big|_{\nu=const.} &\propto \nu^2 \left(\frac{1}{(\tau_x - \tau_{m+1})^2} - \frac{1}{(\tau_x - \tau_m)^2}\right) \\
&\simeq \nu^2 \left(\frac{2}{\tau_x^3}(\tau_{m+1} - \tau_m) + \frac{3}{\tau_x^4}(\tau_{m+1}^2 - \tau_m^2)\right) + \mathcal{O}(\tau_x^{-5}) .
\end{aligned} \tag{D.16}$$

By comparing Eq.(D.15) with Eq.(D.16), we see that in the limit $\tau_x \gg 1$, the gaps between consecutive $m$ at fixed $\nu$ are much smaller that the ones for different $\nu$. See also Fig.12, where we compare these estimates with the exact spectrum obtained solving numerically the boundary condition (D.11). Notice that, in the limit of $m \gg 1$, Eq.(D.16) becomes

$$\Delta h_m\big|_{\nu=const.} \simeq 4\nu^2 \left[\frac{1}{m}\frac{1}{\tau_x^3} + 3\log(2m)\frac{1}{\tau_x^4}\right] + \mathcal{O}(\tau_x^{-5}) \quad \text{for} \quad \tau_x \gg 1 , m \gg 1 , \tag{D.17}$$

namely the term $\propto \tau_x^{-4}$ becomes dominant. This explains the change in the slope in the bottom curve of Fig. 12.

# E  Solution of the quantum particle on the surface with $n$-constant negative curvatures

Consider the following metric

$$ds^2 = \begin{cases} R_1^2 \left( d\tau^2 + \sinh^2 \tau \, d\phi^2 \right) & \text{for} \quad 0 \leq \tau < \tau_1 \\ R_2^2 \left( d\tau^2 + \sinh^2 \tau \, d\phi^2 \right) & \text{for} \quad \tau_1 < \tau < \tau_2 \\ \dots \\ R_i^2 \left( d\tau^2 + \sinh^2 \tau \, d\phi^2 \right) & \text{for} \quad \tau_{i-1} < \tau < \tau_i \\ \dots \\ R_n^2 \left( d\tau^2 + \sinh^2 \tau_n \, d\phi^2 \right) & \text{for} \quad \tau_n < \tau \leq \tau_n + \tau_L \end{cases} . \tag{E.1}$$

It describes a two-dimensional surface with $n-$constant negative curvatures $K_i = -1/R_i^2$ for $\tau < \tau_n$ and a cylinder of radius $R_n \sinh \tau_n$ and length $R_n \tau_L$ for $\tau_n < \tau \leq \tau_n + \tau_L$. We fix $R = R_1 \leq R_2 \leq \dots \leq R_n$, such that $R$ is the smallest scale of the problem. The Laplace-Beltrami operator

$$\nabla^2 = \begin{cases} \left[ \frac{1}{\sinh \tau} \frac{\partial}{\partial \tau} \left( \sinh \tau \frac{\partial}{\partial \tau} \right) + \frac{1}{\sinh^2 \tau} \frac{\partial^2}{\partial \phi^2} \right] & \text{for} \quad 0 < \tau < \tau_1 \\ \dots \\ \left( \frac{R}{R_i} \right)^2 \left[ \frac{1}{\sinh \tau} \frac{\partial}{\partial \tau} \left( \sinh \tau \frac{\partial}{\partial \tau} \right) + \frac{1}{\sinh^2 \tau} \frac{\partial^2}{\partial \phi^2} \right] & \text{for} \quad \tau_{i-1} < \tau < \tau_i \\ \dots \\ \left( \frac{R}{R_n} \right)^2 \left[ \frac{\partial^2}{\partial \tau^2} + \frac{1}{\sinh^2 \tau_n} \frac{\partial^2}{\partial \phi^2} \right] & \text{for} \quad \tau_n < \tau < \tau_n + \tau_L \end{cases} \tag{E.2}$$

acts as the Hamiltonian $\hat{H}_0 = -\frac{\hbar^2}{2\mu R_1^2} \nabla^2$ in the Schrödinger equation. The operator $\nabla^2$ is only a function of the dimensionless variables $\tau, \phi$ and of the parameters $\alpha_i = R_i/R$, hence we are exactly in the situation of Eq.(8) in Sec.4.2. Therefore, we expect the dimensionless Lyapunov exponent to be only a function of $R/\ell_{\mathrm{dB}}$ and $\vec{\alpha}$ [cf. Eq.(31)]. To solve the Schrödinger equation, we start by separating variables

$$\Psi(\tau, \phi) = \frac{e^{im\phi}}{\sqrt{2\pi}} F(\tau) , \qquad m = 0, \pm 1, \pm 2, \dots , \tag{E.3}$$

where the discreteness of $m$ comes from requiring the periodicity in $\phi$ and the function $F(\tau)$ shall be written as a piecewise function $F(\tau) = F_i(\tau)$ for $\tau_{i-1} < \tau < \tau_i$ with $i = 1, \dots n+1$ and $\tau_0 = 0$, $\tau_{n+1} \equiv \tau_n + \tau_L$. The functions $F_i(\tau)$ satisfy

$$\left[ \frac{1}{\sinh \tau} \frac{\partial}{\partial \tau} \left( \sinh \tau \frac{\partial}{\partial \tau} \right) - \frac{m^2}{\sinh^2 \tau} + h_i \right] F_i(\tau) = 0 \quad \text{for} \quad i = 1, \dots, n \tag{E.4}$$

$$\left[ \frac{\partial^2}{\partial \tau^2} - \frac{m^2}{\sinh^2 \tau_n} + h_n \right] F_{n+1}(\tau) = 0 , \tag{E.5}$$

with dimensionless energies

$$h_i = \frac{2\mu R_i^2}{\hbar^2} E , \quad h_i = \left( \frac{R}{R_i} \right)^2 h_1 . \tag{E.6}$$

Eq.(E.4) is the Legendre equation (B.1) for $z = \cosh\tau$, hence for $i = 1,\ldots,n$ the solutions are given by combination of Legendre functions for the first and second order. Eq.(E.5) yields a simple oscillatory behaviour. Therefore, the $k$-th normalized eigenstate $\Phi = g^{1/4}\Psi$ can be written as

$$
\Phi_k(\tau,\phi) = \frac{e^{im_k\phi}}{\mathcal{N}_k} \times
\begin{cases}
R\sqrt{\sinh\tau}\,F_1^k(\tau) & 0 < \tau < \tau_1, \\
\ldots \\
R_i\sqrt{\sinh\tau}\,F_i^k(\tau) & \tau_{i-1} < \tau < \tau_i, \\
\ldots \\
R_n\sqrt{\sinh\tau_n}\,F_{n+1}^k(\tau) & \tau_n < \tau < \tau_n + \tau_L,
\end{cases}
\tag{E.7}
$$

with

$$
F_i^k(\tau) = c_i P_m^{\ell_i^k}(\cosh\tau) + d_i Q_m^{\ell_i^k}(\cosh\tau), \tag{E.8a}
$$
$$
F_{n+1}^k(\tau) = c_L \cos[\kappa^k(\tau - \tau_n)] + d_L \sin[\kappa_k(\tau - \tau_n)], \tag{E.8b}
$$

where we identified

$$
\ell_i^k(\ell_i^k + 1) \equiv -h_i^k \quad \to \quad \ell_i^k = -\frac{1}{2} + i\sqrt{h_i^k - \frac{1}{4}}, \tag{E.9a}
$$

$$
\kappa_k^2 \equiv h_n^k - \frac{m_k^2}{\sinh^2\tau_n^2}. \tag{E.9b}
$$

The coefficients $c_i, d_i$ as well as $c_L, d_L$ are determined by imposing on each $\tau_i$ the continuity of the normalized wave function (E.7) and of its derivative. First of all, the Legendre function of second kind $Q_m^\ell(z)$ diverges for $z \to 1$, therefore we directly set $c_1 = 1$ and $d_1 = 0$. Using the recursive relation for the derivatives (B.5), a straightforward calculation yields for $i \le n$

$$
c_i = \frac{R_{i-1}}{R_i} \frac{c_{i-1}\left(P_m^{\ell_{i-1}}Q_{m+1}^{\ell_i} - Q_m^{\ell_i}P_{m+1}^{\ell_{i-1}}\right) + d_{i-1}\left(Q_m^{\ell_{i-1}}Q_{m+1}^{\ell_i} - Q_m^{\ell_i}Q_{m+1}^{\ell_{i-1}}\right)}{Q_m^{\ell_{i-1}}P_m^{\ell_i} - Q_m^{\ell_i}P_{m+1}^{\ell_i}}, \tag{E.10a}
$$

$$
d_i = \frac{R_{i-1}}{R_i} \frac{c_{i-1}\left(P_{m+1}^{\ell_{i-1}}P_m^{\ell_i} - P_m^{\ell_{i-1}}P_{m+1}^{\ell_i}\right) + d_{i-1}\left(Q_{m+1}^{\ell_{i-1}}P_m^{\ell_i} - Q_m^{\ell_{i-1}}P_{m+1}^{\ell_i}\right)}{Q_m^{\ell_{i-1}}P_m^{\ell_i} - Q_m^{\ell_i}P_{m+1}^{\ell_i}}. \tag{E.10b}
$$

On the other hand for $\kappa \neq 0$, one finds

$$
c_L = c_n P_m^{\ell_n} + d_n Q_m^{\ell_n}, \tag{E.11}
$$
$$
\kappa\, d_L = c_n P_{m+1}^{\ell_n} + d_n Q_{m+1}^{\ell_n} + \left(m + \frac{1}{2}\right)\coth\tau_n\left(c_n P_m^{\ell_n} + d_n Q_m^{\ell_n}\right),
$$

where $P_m^{\ell_i} = P_m^{\ell_i}(\cosh\tau_i)$ and $Q_m^{\ell_i} = Q_m^{\ell_i}(\cosh\tau_i)$ for all $m$. The quantization condition determining the $k$-th eigenvalue is fixed by imposing the wave-function to be zero at the end of the cylinder for $\tau = \tau_L$, this leads to

$$
\tan[\kappa_k \tau_L] = -\frac{c_L}{d_L}. \tag{E.12}
$$

Notice that the nature of the wave-function changes (from oscillatory to exponentially decaying), as soon as the argument inside the square-root of Eq.(E.9a) changes sign. See Fig.7 in the main text as an illustrative example in the case $n = 2$. This immediately leads to the gaps $\Delta_i = \frac{1}{4}\frac{\hbar^2}{2\mu R_i^2} = \frac{\hbar^2}{2\mu R^2}\delta_i$ with $\delta_i = \frac{1}{4}\frac{R^2}{R_i^2}$ [cf. Eq.(63) in the main text].

# F  Two quantum mechanisms, more generally

## F.1  Avoidance of curved regions

In Section 5.5, we have focused on the surface of constant negative curvature. Let us generalize our findings to generic two-dimensional surfaces. The metric can be generally written in geodesic polar coordinates as [39]

$$ds^2 = R^2 \left[ d\tau^2 + g(\tau, \phi) d\phi^2 \right] , \tag{F.1}$$

where $R$ is length characterizing the surface, $R\tau$ is the geodesic length and $\phi$ is the angle. In this notation $\overline{g} = R^4 g(\tau, \phi)$ is the determinant of the metric. The Gaussian curvature $K$ can be computed as [39]

$$K = -\frac{1}{R^2} \frac{1}{\sqrt{g}} \frac{\partial^2 \sqrt{g}}{\partial \tau^2} . \tag{F.2}$$

It is known that quantum mechanics on manifolds can give rise to potentials depending the extrinsic curvature, see e.g. Refs. [40, 41]. Here, we will show how to derive such potentials for the two-dimensional metric in terms of the intrinsic curvature [14]. As we have discussed at the beginning of the section within a semi-semi classical approach, the presence of such potential implements the bound to chaos.

The Laplace-Beltrami operator (13) for the geodesic coordinates (F.1) on a surface with *constant* curvature reads

$$\nabla^2 = \frac{1}{R^2 \sqrt{g}} \frac{\partial}{\partial \tau} \left( \sqrt{g} \frac{\partial}{\partial \tau} \right) + \frac{1}{g} \frac{\partial^2}{R^2 \partial \phi^2} = \frac{1}{R^2} \frac{\partial^2}{\partial \tau^2} + \frac{1}{R^2 g(\tau)} \frac{\partial^2}{\partial \phi^2} + \frac{\Gamma_\tau}{R^2} \frac{\partial}{\partial \tau} , \tag{F.3}$$

where we have defined

$$\Gamma_\tau \equiv \frac{1}{2} \frac{g'}{g} = \frac{1}{2} \frac{\partial}{\partial \tau} \log g , \tag{F.4}$$

or equivalently $\sqrt{g} = e^{\int^\tau \Gamma_t dt}$. We apply the Schrödinger equation (12) to $\Psi(\tau, \phi) = g^{-1/4} \Phi(\tau, \phi)$, obtaining

$$-\frac{\hbar^2}{2\mu R^2} \left( \frac{\partial^2}{\partial^2 \tau} + \frac{1}{g(\tau)} \frac{\partial^2}{\partial \phi^2} \right) \Phi + V_{\text{eff}}(g) \Phi = E \Phi , \tag{F.5}$$

where we have defined the effective potential, as

$$V_{\text{eff}}(g) \equiv \frac{\hbar^2}{8\mu R^2} \left[ \Gamma_\tau^2 + 2\Gamma_\tau' \right] = \frac{\hbar^2}{2\mu R^2} g^{-1/4} \frac{\partial^2 g^{1/4}}{\partial^2 \tau} = \frac{\hbar^2}{2\mu} \left[ -\frac{K}{4} + \frac{1}{8R^2} \frac{\partial^2 \log g}{\partial \tau^2} \right] , \tag{F.6}$$

where $K$ is the Gaussian curvature in Eq.(F.2). Eq.(F.6) is obtained by comparing it with the definition (F.2). $V_{\text{eff}}(g)$ vanishes in the case of flat surfaces ($K = 0$ and $g = r^2$), it is attractive in the case of positive curved regions $K > 0$, while it is repulsive for negative curvature $K < 0$. Remarkably, is the presence of negative curvature – classically the origin of chaotic behaviour – that at the quantum level generates the repulsive potential [cf. Eq.(50)] implementing the bound.

---

[14]Extrinsic geometry concerns properties of a surface in relation to the embedding on a three dimensional space. Intrinsic properties of surfaces are properties that can be measured within the surface itself without any reference to a larger space.

## F.2 Spreading of the wave-packet

Let us discuss the simplest instance in which quantum fluctuations modify the classical exponential instability on the pseudosphere: We consider a Gaussian wave-packet evolving on the classical geodesic and we study how its spreading modifies the Lyapunov regime. To highlight the salient features, we focus here on a finite (yet big) portion of the surface with constant negative curvature (36) for $\tau \leq \tau_x \gg 1$. See App. D for a detailed study of the classical and quantum dynamics of this model.

On the surface of constant negative curvature, the classical Lyapunov exponent comes from the geometry, being distances computed via the non-trivial metric. By choosing two initial conditions with the same radial coordinate $\tau_0$ and two angles differing by $\delta\phi(0)$, the spatial separation between two geodesics is given by

$$
\begin{aligned}
\left(\frac{ds(t)}{ds(0)}\right)^2 &= \sinh^2 \tau(t)\left(\frac{d\phi(t)}{d\phi(0)}\right)^2 \\
&= (\sinh \tau(t)\{\phi(t), I(0)\})^2 \sim \sinh^2 \tau(t) \sim e^{2\lambda_c t} \quad \text{for} \quad t \to \infty, \quad (\text{F.7})
\end{aligned}
$$

where in the second line we have use the definition of the Poisson Brackets, to emphasise analogy with the square-commutator (see below). In the third line we have simply used that angular variations remain almost constant in time, see App. D.1.

In order to understand how geometric chaos translates into quantum mechanics, we consider as square commutator as the quantum version of Eq.(F.7), i.e.

$$
c(t) = -\langle \sinh^2 \hat{\tau}(t)[\hat{\phi}(t), \hat{I}(0)]^2\rangle. \quad (\text{F.8})
$$

If we evaluate the expectation value over a factorized wave-packet in the angular and radial coordinates we can approximate the square commutator with

$$
c(t) \simeq \langle \sinh^2 \hat{\tau}(t)/R\rangle \sim \langle\Phi(t)|e^{2\hat{\tau}/R}|\Phi(t)\rangle = \int d\tau\, |\Phi(\tau,t)|^2 e^{2\tau/R}, \quad (\text{F.9})
$$

where $\tau$ is now a dimensional variable and $\Phi(\tau, t)$ is the time-dependent wave-function solution of the Schrödinger equation. We consider the system initialized in a Gaussian wave-packet with variance $\sigma_0$ and centered around $\tau_0$, that can be set to zero $\tau_0 = 0$ without loss of generality. The value of $\sigma_0$ is determined by maximizing the uncertainty principle, i.e. $\sigma_0 = \frac{\hbar}{2\Delta p}$. The wave-packet undergoes free evolution and at time $t$ one has [42]

$$
|\Phi(\tau, t)|^2 = \frac{2}{\sigma(t)\sqrt{2\pi}} e^{-\frac{(\tau - vt)^2}{2\sigma(t)^2}}, \quad (\text{F.10})
$$

where $v = p/\mu$ is the velocity and $\sigma(t) = \sigma_0\sqrt{1 + \left(\frac{t}{T_N}\right)^2}$ is the standard spreading of the variance of the wave-packet with $T_N = \frac{2\mu\sigma_0^2}{\hbar}$ usually referred as the natural time [42]. In this case, the Ehrenfest time [cf. Eq.(33)] $T_{\text{Ehr}} \sim \log \text{Vol} \sim \tau_x$ can be made arbitrarely large increasing $\tau_x$. A straightforward calculation of the Gaussian integral in Eq.(F.9) leads to the result of Eq.(59) and to the relative conditions on the extent of the wave-packet and on the de Broglie length [cf. Eqs.(60)-(61)].

We are now in position to generalize these arguments. Here, we have shown that on a two-dimensional manifold, the Lyapunov exponent can be defined only for $p/\Delta p \gg 1$ and $R/\ell_{\text{dB}} \gg \frac{1}{4\pi}$, being $p$ and $(\Delta p)$ the momentum (spreading) of the radial coordinate. Below these length scales, the Lyapunov exponential growth is suppressed by a constant or superexponential regime.

On generic manifolds, nearby geodesics separate exponentially with the path length $\ell$ as $\Delta(\ell) \sim \Delta_0 e^{\ell/s}$ [cf. Eq.(23)] with $s$ the geodesic separation. In $N$ dimensions, it scales with $N$ as $s = R\sqrt{N}$, with $R$ the characteristic length, finite in the thermodynamic limit [9]. This ensures the proper scaling of the Lyapunov exponent $\lambda_c = \frac{p_i}{\mu R} = \frac{p}{\mu s} = \mathcal{O}(1)$ in the thermodynamic limit. Here $p \sim \sqrt{\mu N T}$ is the total momentum, while $p_i \sim \sqrt{\mu T}$ the one of the single degree of freedom. To study the impact of quantum fluctuations one can evaluate the average of $\Delta(\ell)$ over the probability of being on $\ell$ at time $t$, i.e.

$$\langle \Delta(\ell) \rangle_t = \Delta_0 \int d\ell \, P(\ell, t) e^{\ell/s} \,, \tag{F.11}$$

being $P(\ell, t) = |\Phi(\ell, t)|^2$ the modulus square of the wave-packet at time $t$. Notice that the exponential factor in the integrand makes Eq.(F.11) like a saddle point integral and therefore the result will be dominated by largest values $\ell$ at time $t$. The initial state is chosen as a Gaussian wave-packet factorized in all the directions $\{x_i\}_{i=1,\dots,N}$. The initial variance $\sigma_0 = \hbar/2\Delta p_i$ maximizes the uncertainty principle on each direction. Then, assuming the wave-packet remains factorized, at time $t$ one has

$$P(\ell, t) = P(\{x_i\}, t) \sim \prod_{i=1}^{N} e^{-\frac{(x_i - p_i t/\mu)^2}{2\sigma(t)^2}} \,, \tag{F.12}$$

with $p_i$ the conjugate variable to $x_i$ and $\sigma(t)^2 = \sigma_0^2(1 + \left(\frac{\hbar}{2\mu\sigma_0^2}\right)^2 t^2)$ representing the spreading of the uncertainty of the wave-function at time $t$. One can estimate $\ell \sim x_i \sqrt{N}$ and, by completing the squares in the integral (F.11), one obtains

$$\langle \Delta(\ell) \rangle_t = \exp\left[ \frac{\sigma_0^2}{2R^2} + \lambda_c t + \frac{1}{2}\left(\frac{\Delta p_i}{p_i}\right)^2 \lambda_c^2 t^2 \right] \,, \tag{F.13}$$

where we have used the definition of the temporal Lyapunov exponent $\lambda_c = p/s\mu$. This expression has exactly the same scaling as the one of the two-dimensional surface, so that the problem is not "cured" by the large $N$ limit.

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
