# Peer review of "Low temperature quantum bounds on simple models"

_SciPost Physics, doi:SciPost Phys. 13, 006 (2022)_

## Round 2 · Referee Report · Anonymous (Referee 1) · 2022-2-26

Strengths

  1. The paper appears to contain some interesting numerical results for the classical and quantum billiard problem on a pseudosphere.

Weaknesses

  1. The paper is not readable.

Report

I do not recommend publication. While it is possible that the results are significant and interesting, the work is not written in a way that clearly conveys what was done.

As far as I understand it, the authors considered quantum and classical billiard problems involving a pseudosphere. They look at the pseudosphere glued to a cylinder, the pseudosphere cut-off with a hard-wall, and what I would call a "banded'' pseudosphere where rings of pseudosphere with different radii of curvature are glued together. Furthermore, they compute analogs for these systems of the out of time ordered correlation (OTOC) function of [4]; importantly for this OTOC there is a bound on its exponential behavior which has been argued to be a bound on chaos. I am not 100% certain, but I believe Kurchan and Pappalardi's central thesis in this paper is that this bound comes closest to being saturated in their models at the cross-over between quantum and classical behavior. I am not certain because I have not been able to pinpoint where in the draft they actually show and/or argue for the details of this thesis.

My problems with the manuscript start with the abstract, which instead of conveying the details of what is done, fail to mention the pseudosphere.
Given the juxtaposition of "we consider free motion on curved space" with "Thermodynamic examples of these include quantum hard-spheres and quantum spin liquids," one might be forgiven for assuming the authors will consider quantum hard-spheres and quantum spin liquids in detail.
Figure 1 and the brief discussion in section 7 instead present only some history and speculation about these topics.
Further, to discuss saturating the bound at $T=0$ I thought was misleading as well. I think the authors mean saturate in the low temperature limit, or perhaps more accurately low energy limit since they never really work with a thermal ensemble and instead assume equipartition $E = NT/2$.

I found the following two sentences ludicrous: "This paper aims at being pedagogical. For this reason, all the details of the calculations are reported in the appendix." I found the only pedagogical material to be in the appendices. The main part of the paper was very difficult to follow. Often quantities were introduced which I only later found were defined in the appendices, for example $N_{cut}$ and $m_n$.

Requested changes

1) (1) is not actually a bound. It is known to be violated in certain circumstances.

2) "On the other hand, the bound to chaos – depending only on temperature and the Planck constant – corresponds to a genuine quantum effect.'' What is the "bound to chaos''?

3) "The goal of this work'' should be indented.

4) What does it mean for a bound to be "effective non-trivially''?

5) "The first is that $\hbar / T$ be finite, which holds in the semiclassical limit for very low temperatures corresponding to the lowest classical energies of the system.'' Surely this is $\hbar/T$ is finite provided $T>0$?

6) "In the presence of ground-state degeneracies or quasi-degeneracies at the lowest energies; the system may instead be non-harmonic (and chaotic) even "at the bottom of the well'' ''$\to$ "In the presence of ground-state degeneracies, or quasi-degeneracies at the lowest energies, the system may instead be non-harmonic (and chaotic) even "at the bottom of the well'' ''

7) "In Sec. 6 we discuss how the bound holds in the limit of zero temperature in presence of a hierarchy of length scales'' Which bound? The authors need to make it clearer they focus only on (2).

8) The authors claim to find a violation of the bound (2) at the end of section 2 and again on pp 19-20 where the discrepancy is attributed to "regulation'' issues. I am upset with such a light treatment of a potentially significant result. It seems that this entire manuscript was built around comparing their results to the bound in (2). If they are comparing apples and oranges, then I need to understand in greater detail the map from apples to oranges. If they are comparing apples to apples, then I need to know why (2) is wrong or this manuscript is wrong.

9) "reflecting wall at $\tau = \tau_x$'' $\to$ "reflecting wall at $\tau = \tau_L$''

10) On p 14, the authors claim the high energy particles spend most of their time on the cylinder and then later assume ergodicity, which seems mutually inconsistent.

11) At the top of p 15, the authors refer to the Loschmidt echo, which I thought was a purely quantum phenomenon, in a section on the classical behavior of their billiard. I found this confusing.

12) In (49), I did not understand on what the minus sign on the far left was supposed to act, and (50) looked potentially unbounded below.

13) In gluing the cylinder to the pseudosphere, one could get rid of the step function by adding a constant potential to the cylinder. Did the authors consider this possibility?

14) In figure 6a, it looks like the data is growing a bit faster than the green line, i.e. violating the bound?

15) "In this case, the repulsive potential $\delta$ is absent, as for a finite portion of the hyperboloid with a wall at some $\tau_x$.'' I think the authors mean to introduce the example of a pseudosphere with a hard wall, but it's not clear from the text.

16) "In the second line, we have used...'' but there is no second line in the preceding equation

17) On p 38, there is an "Eq.(??),"

18) The list of small issues that I found is longer than my patience to describe them. I'll conclude by saying that there were a number of very creative and mutually inconsistent spellings of pseudosphere, Schr\"odinger, Loschmidt and dimensional.

---

## Round 2 · Referee Report · Anonymous (Referee 2) · 2022-3-15

Strengths

1- The paper deals with bounds on transport and chaos, which have attracted a lot of attention in recent years across different communities 2- It relates these bounds to the fundamental concepts of quantum mechanics, such as the uncertainty principle 3- It adds to the existing literature by offering a pedagogical and intuitive insight on how the bounds arise at low temperatures in systems of free particles propagating in the curved space-time

Report

This work considers how the bounds on chaos and transport emerge at low temperatures.It is shown that classically the bounds are violated at low temperatures for free particles on curved manifolds. Different quantum mechanisms that restore the bounds are identified, as well as a potential scenario where the bound on chaos is violated.

The paper is considering a timely topic of broad interest, offering new insights. The presentation is clear and pedagogical. Therefore I suggest the publication of the paper in SciPost physics after a small revision addressing the points below (point 3 in particular).

Requested changes

1- For completeness it would be useful to write what N in the large N limit is 2- In equation 14b the expression should be divided by the partition sum 3- If I am not mistaken, the Helfand functions should correspond to the time integral of the current divided by sqrt[VT] instead of the time derivative, in order for expressions 14 and 15 to match. Is this accounted for in other results? 4- I would suggest using some other letter instead of s in equation 23, which was already used as the entropy density 5- There are some typos: boundS at T=0 in abstract, useD the definition on page 32 From this expresion IT is clear on page 32 missing equation reference on page 38 Missing reference in ref [23]

---

## Round 5 · Referee Report · Anonymous (Referee 1) · 2022-5-3

Weaknesses

I remain puzzled by mechanisms (2) and (3) in the draft. Regarding (2), the existence of a repulsive potential in their toy model seems like a choice of how they glue the cylinder to the pseudosphere, rather than a generic feature of these systems. The claim that (3) leads to super-exponential behavior and a violation of the quantum Lyapunov bound is troubling, given that there exists a proof of the Lyapunov bound. Could it be that the proof assumes a flat space-time, and the pseudosphere is curved?

Report

The authors clearly spent a great deal of time addressing my earlier concerns, and the manuscript looks vastly improved. I can read it much more easily now.

---

## Round 5 · Author Response

Dear Editor,

thank you for handling our submission.

We are pleased to thank the referees, whose insightful comments and observations helped us further improve and clarify our work. In this resubmission, we believe to have addressed all the points raised by the Referees. We append below a detailed response to the reports and the list of changes.

Yours sincerely,

Silvia Pappalardi and Jorge Kurchan

Reply to the Referee 1

We thank the Referee for their careful reading of our manuscript and for a number of comments and suggestions. Following the referee's remarks, we have written a new abstract, and completely changed the introduction and the main parts of the manuscript. We have also performed a careful rewriting of the paper.

All the comments of the referee are addressed in detail below and have been used to change accordingly the manuscript. There are points that we did not address because they are beyond the scope of this paper: the Referee does not believe in the bound on viscosity [Eq. (1) in the main text]; while we are aware that there has been a debate on this bound, testing its validity goes well beyond our goals here. The current version of the manuscript has improved a lot in its clarity and we believe it is now suitable for publication in Scipost Physics.

My problems with the manuscript start with the abstract, which instead of conveying the details of what is done, fail to mention the pseudosphere. Given the juxtaposition of "we consider free motion on curved space" with "Thermodynamic examples of these include quantum hard-spheres and quantum spin liquids," one might be forgiven for assuming the authors will consider quantum hard-spheres and quantum spin liquids in detail. Figure 1 and the brief discussion in section 7 instead present only some history and speculation about these topics.

We thank the referee for this comment. We have rewritten the abstract including the details of the toy model studied in the paper. Furthermore, we have specified in the introduction that hard spheres and quantum spin liquids are only a motivation and moved Figure 1 and the discussion to Section 7.

Further, to discuss saturating the bound at T=0 I thought was misleading as well. I think the authors mean saturate in the low temperature limit, or perhaps more accurately low energy limit since they never really work with a thermal ensemble and instead assume equipartition E=NT/2.

Following the referee's comment, we have indeed changed the $T=0$ with "in the low-temperature limit'' or "at the lowest temperature" or "at the lowest energies" both in the abstract and the text. However, we would like to remark that all the general quantum scalings of Section 4 are indeed derived for thermal ensembles.

The main part of the paper was very difficult to follow. Often quantities were introduced which I only later found were defined in the appendices, for example Ncut and mn.

We thank the referee for pointing out these typos. We have performed a careful rewriting of the text and specified some of the details (that were before in the Appendix) in the main part of the manuscript. For instance, we have moved the exact eigenstates of the toy model from the Appendix to Section 5.4.

(1) is not actually a bound. It is known to be violated in certain circumstances.

We are aware that there is no general consensus on the status of the quantum bound on the viscosity. However, addressing its validity is beyond the scope of this paper. We have added to the introduction the following sentence "Even if there is no general consensus on the bounds on transport, it was argued that they are related to the emergence of a Planckian time-scale $\tau_{Pl}= \hbar / T$, depending only on the Planck constant $\hbar$ and on the absolute temperature $T$ in units of energy."

"On the other hand, the bound to chaos – depending only on temperature and the Planck constant – corresponds to a genuine quantum effect.'' What is the "bound to chaos''?

The bound on the Lyapunov exponent in Eq.(2) is known as "the (quantum) bound to chaos". We have added this sentence right after the equation to make it more clear.

"The goal of this work'' should be indented.

We have completely rewritten the abstract and the introduction of the paper to make clear that the goal of the paper is to understand the physical mechanisms underlying the bound to chaos.

What does it mean for a bound to be "effective non-trivially''?

A trivial way to satisfy the bound to chaos is to have, for instance, a vanishing Lyapunov exponent. This is the case of the low-temperature semi-classical limit of a system with an isolated minimum: quantum fluctuations would just result in oscillations around the classical ground state because of the linearization of the dynamics, with zero Lyapunov exponent. Following the referee's comment, we have re-written the part of the introduction.

"The first is that $\hbar/T$ be finite, which holds in the semiclassical limit for very low temperatures corresponding to the lowest classical energies of the system.'' Surely this is $\hbar/T$ is finite provided $T>0$?

Sure, we have added in the text ``for $T>0$''.

"In the presence of ground-state degeneracies or quasi-degeneracies at the lowest energies; the system may instead be non-harmonic (and chaotic) even "at the bottom of the well'' '' "In the presence of ground-state degeneracies, or quasi-degeneracies at the lowest energies, the system may instead be non-harmonic (and chaotic) even "at the bottom of the well'' ''.

We have changed the manuscript accordingly.

"In Sec. 6 we discuss how the bound holds in the limit of zero temperature in presence of a hierarchy of length scales'' Which bound? The authors need to make it clearer they focus only on (2).

We thank the referee for this comment, we have specified there (and in other parts of the manuscript) that the discussion on the mechanisms only concern the bound to chaos in Eq.(2).

The authors claim to find a violation of the bound (2) at the end of section 2 and again on pp 19-20 where the discrepancy is attributed to "regulation'' issues. I am upset with such a light treatment of a potentially significant result.

We indeed found the mechanism (3) puzzling: we expected quantum effects to suppress the Lyapunov, rather than enhancing it with a super-exponential growth. For this reason, we attributed this discrepancy to possible regulations issues. Actually, the spreading of the wave-packet makes the Lyapunov regime ill-defined and hence the applicability of the bound in (2). We have changed the manuscript accordingly. We thank the referee for this comment and we agree that understanding the role of the regularization onto the bound to chaos is an interesting problem that goes beyond what is addressed in this paper.

"reflecting wall at $\tau = \tau_x$. "reflecting wall at $\tau = \tau_L$''

We have corrected the typo.

On p 14, the authors claim the high energy particles spend most of their time on the cylinder and then later assume ergodicity, which seems mutually inconsistent.

The equilibrium distribution of the system is indeed ergodic, in the sense that position and momenta are uniformly distributed, but according to the proper metric of the model. This results in the fact that the particle spends more time at large radial coordinates, both classically and for high-energy eigenstates. This fact has been illustrated in detail in Appendix A and Figure 9, where we show the radial equilibrium distribution. To make it more clear in the text we have added the following comment in the text "Note that, once integrability is broken (see below), this corresponds to an ergodic equilibrium distribution that is uniform with respect to the metric, see Fig. 4 in the Appendix.". This is taken into account in the evaluation of the total classical Lyapunov exponent: the pseudosphere value $\lambda_c$ is multiplied by the ratio of volume spent in the pseudosphere over the total one.

At the top of p 15, the authors refer to the Loschmidt echo, which I thought was a purely quantum phenomenon, in a section on the classical behavior of their billiard. I found this confusing.

We meant that, even if the system is integrable, the geodesics are exponentially unstable. Therefore the smallest perturbation would destroy the integrability and induce chaos with a Lyapunov exponent which is independent of the perturbation strength. This is the scenario that underlies Loschmidt echo experiments, which are well defined in both classical and quantum dynamics, see e.g. [Veble and Prosen Phys. Rev. E 72, 025202(R) (2005)] or [Fine, Elsayed, Kropf, and de Wijn, Phys. Rev. E 89, 012923 (2018)]. In order to avoid confusion for the reader, we have substituted the reference to the Loschmidt echo protocols and explained the concept of exponential instability.

In (49), I did not understand on what the minus sign on the far left was supposed to act, and (50) looked potentially unbounded below.

We thank the referee for the comment, Eq.(49) was indeed confusing. We have re-written it to make it more clear. The potential divergence $\propto 1/\sinh^2 \tau$ has no effects in our setting. It is only attractive for the zero-angular momentum subspace. For this case, it is well known that potential we have studied in detail what effect this attraction could possibly have and found that it was not important enough to make a discussion worth while here, since we are interested in higher energy levels. We have added an interesting reference on this point [H. E. Camblong, L. N. Epele, H. Fanchiotti and C. A. G. Canal, Renormalization of the inverse square potential, PRL 85(8), 1590 (2000).].

In gluing the cylinder to the pseudosphere, one could get rid of the step function by adding a constant potential to the cylinder. Did the authors consider this possibility?

We thank the referee for the relevant comment. This issue is exactly what drove us to identify the mechanism (3). In fact, the gap induced by the curvature (which implements the bound) could always be balanced by adding a potential on the cylinder that favours visiting the most curved part. This situation is indeed equivalent to studying what happens in the presence of only one radius of curvature wit a wall instead of the cylinder. This precisely led us to study the pseudosphere with a hard wall at a given distance, which is the model studied in Section 5.6 and in Appendix D. We have re-written the beginning of Section 5.6 to emphasize this point.

In figure 6a, it looks like the data is growing a bit faster than the green line, i.e. violating the bound?

We believe that the growth rate of Fig.6a is compatible with the classical Lyapunov $\lambda_c$. Small deviations are in fact expected at small times, see for instanc Ref.[E. Rozenbaum, S. Ganeshan and V. Galitski, PRL 086801, 2017] or Ref.[Lerose and Pappalardi, PRA 102, 2022].

"In this case, the repulsive potential $\delta$ is absent, as for a finite portion of the hyperboloid with a wall at some $\tau_x$.'' I think the authors mean to introduce the example of a pseudosphere with a hard wall, but it's not clear from the text.

Following this suggestion, we have written explicitly "quantum solution of a free particle on a pseudosphere with an hard wall at some $\tau_x$".

"In the second line, we have used...'' but there is no second line in the preceding equation.

We have corrected the error.

On p 38, there is an "Eq.(??),"

We have added the reference.

I'll conclude by saying that there were a number of very creative and mutually inconsistent spellings of pseudosphere, Schr\"odinger, Loschmidt and dimensional.

We have corrected all the misspellings we could find.

Reply to the Referee 2

We thank the referee for their reading of the paper and support in suggesting publication in SciPost Physics after a small revision. In the new version of the manuscript, we have performed a careful rewriting and addressed all the issues specified by the referee, as we comment below.

  1. We have changed the sentence are defined only in the large $N$ limit'' toare defined only for systems of $N$ elementary constituents (such as spins or fermionic sites) with all-to-all interactions in the large $N$ limit''.
  2. Changed.
  3. We thank the referee for pointing out this typo that we have corrected.
  4. we have changed the notation $s\to S$ to the volume density of entropy in Eq.(1). We wanted to keep $s$ for the characteristic path length in Eq.(23), which is the standard notation, see e.g. Ref.[10] of the manuscript.
  5. Abstract changed; changed; changed; changed; reference added.

---

## Round 5 · List of Changes

• we have completely rewritten the abstract;
  • we have completely rewritten the introduction;
  • we have moved the first figure (with the examples of the macroscopic systems) to Section 7. We have also specified that we consider it as only a motivation;
  • we have indeed changed the $T=0$ with in the low-temperature limit'' orat the lowest temperatures'' or ``at the lowest energies'';
  • we have moved the exact solution of the toy model from the Appendix to the main text (now Eqs.(52-55));
  • we have specified that there is no consensus on the bound on the viscosity;
  • we have simplified the discussion of the rescaling of the dimensionless quantities;
  • we have specified that Eq.(2) is "the quantum bound to chaos";
  • we have changed the discussion about the mechanism (3) and the reference to possible "regulation issues";
  • we have explained better the ergodic distribution in the curved metric;
  • we have removed the reference to the Loschmidt echo and substituted it with the description of instability;
  • we have corrected a series of typos and misspellings;
  • added the partition function in Eq.(14b);
  • added what is $N$ in the large $N$ limit;
  • corrected the definition of the Helfland moments;
  • we have changed the notation $s\to S$ to the volume density of entropy in Eq.(1). We wanted to keep $s$ for the characteristic path length in Eq.(23), which is the standard notation, see e.g. Arnold.
  • added reference in [23].

---

## Editorial Decision

published